# Dispersal from the Qinghai-Tibet plateau by a high-altitude butterfly is associated with rapid expansion and reorganization of its genome

Youjie Zhao [1,2,6], Chengyong Su [1,6], Bo He [1,6], Ruie Nie [1], Yunliang Wang [1], Junye Ma [3], Jingyu Song [4], Qun Yang [3,5] ✉ & Jiasheng Hao [1] ✉

*Parnassius glacialis* is a typical "Out of the QTP" alpine butterfly that originated on the Qinghai-Tibet Plateau (QTP) and dispersed into relatively low-altitude mountainous. Here we assemble a chromosome-level genome of *P. glacialis* and resequence 9 populations in order to explore the genome evolution and local adaptation of this species. These results indicated that the rapid accumulation and slow unequal recombination of transposable elements (TEs) contributed to the formation of its large genome. Several ribosomal gene families showed extensive expansion and selective evolution through transposon-mediated processed pseudogenes. Additionally, massive structural variations (SVs) of TEs affected the genetic differentiation of low-altitude populations. These low-altitude populations might have experienced a genetic bottleneck in the past and harbor genes with selective signatures which may be responsible for the potential adaptation to low-altitude environments. These results provide a foundation for understanding genome evolution and local adaptation for "Out of the QTP" of *P. glacialis*.

The rapid uplift and climate changes of the Qinghai-Tibet Plateau (QTP) have significantly influenced modern biological diversification since the early Cenozoic era [1–4]. Habitat fragmentation and topographical isolation caused by mountain uplift have resulted in local speciation or rapid adaptation of some organisms [4–6]. A few species have been found to spread out of the QTP region driven by intense climate changes [7,8]. However, these species could experience different fates after a long-term evolutionary process as they encountered various challenges arising from low-altitude environments (such as oxidative damage, higher atmospheric pressure, new competitors or predators, and invasion of pathogenic microorganism in warmer

regions). Large mammals generally underwent severe extinction or population decline [9], while the small and medium-sized mammal groups, such as rodents, bats, lagomorphs, and insectivores, experienced rapid radiation and range expansion [10–12]. Unfortunately, little is known about the drivers of evolution and adaption for these organisms.

*Parnassius* is a genus of alpine butterflies belonging to the group Papilionidae (Lepidoptera), including about 55 described extant species [13]. Previous studies have shown that *Parnassius* butterflies have the largest genome among all members of Papilionidae [14]. The reported genome sizes for *P. orleans* [15] and *P. apollo* [16] are up to 1.23 GB and 1.39

[1]College of Life Sciences, Anhui Normal University, Wuhu 241000, China. [2]College of Big Data and Intelligent Engineering, Southwest Forestry University, Kunming, 650224 Yunnan, China. [3]State Key Laboratory of Palaeobiology and Stratigraphy, Center for Excellence in Life and Palaeoenvironment, Nanjing Institute of Geology and Paleontology, Chinese Academy of Sciences, Nanjing 210008, China. [4]College of Animal Science, Shandong Agricultural University, Taian 271000, China. [5]Nanjing College, University of Chinese Academy of Sciences, Nanjing 211135, China. [6]These authors contributed equally: Youjie Zhao, Chengyong Su, Bo He. ✉e-mail: qunyang@nigpas.ac.cn; jshaonigpas@sina.com

GB, respectively, which are approximately 3.4–5.7 times larger than the genome sizes reported for the *Papilio xuthus* butterfly (244 Mb)[17]. The phylogeny and biogeographic history have revealed that the ancestor of *Parnassius* probably originated from the high mountains of central Asia to west China around 17–14 million years ago (Ma) and rapidly diverged along with the uplift of the QTP and climate changes, following the successive dispersal to Europe and North America[5,13,18]. The extant *Parnassius* butterflies in China are found to have originated from the QTP region, and most of these species diversified in situ (e.g., *Parnassius acdestis*, *P. simo*, *P. orleans*, *P. cephalus*, and *P. epaphus*, etc.). The remaining few species are dispersed into high-latitude regions (e.g., *P. apollo* and *P. nomion* in Xinjiang and Northeast China), or low-altitude and low-latitude regions (e.g., *Parnassius glacialis* in East China).

To the best of our knowledge, *P. glacialis* is the only known species in the genus that dispersed eastwards from the QTP region to southern China (south of Yangtze River) with estimated dates coincided closely with the Kunlun-Huanghe tectonic movement occurring from 1.1 to 0.6 Ma[18]. This species mostly inhabits in mountainous areas at altitudes of 300 to 2,000 m above sea level (a.s.l.). Previous studies on genotyping-by-sequencing (GBS) have suggested that a few low-altitude populations of China have diverged into a separate clade[18]. Meanwhile, the reported morphological data indicated that *P. glacialis* butterflies dwelling in low-altitude areas are larger in body size than those in relatively high-altitude regions[19]. Studies have shown that the altitudinal body size cline may be related to oxygen level, temperature, climate variation, and seasonality[20–23]. Although different selective pressures presumably exist for populations at different altitudes along with varied body size, the potential mechanisms of evolution and adaption are still poorly understood due to the lack of *P. glacialis* genome resources.

In this work, we assessed the genome size variation among six *Parnassius* species at different altitudes. Subsequently, we assembled the chromosome-level genome of *P. glacialis* and explored the role of transposable elements (TEs) in the evolution of its large genome through comparative analysis with the reported genomes of *P. orleans*[15] and *P. apollo*[16]. Based on the genome sequencing of 41 individuals from 9 *P. glacialis* populations at different altitudes ranging from 300 to 1800 m a.s.l., we analyzed the genetic structure and explored the impact of TEs on the genetic differentiation for these populations. This study will help us to understand the mechanisms of genome evolution and local adaptation for *P. glacialis* butterflies.

## Results

### Genome survey and assembly

In order to explore the relationship between genome size and elevation for genus *Parnassius*, genome sizes of six representative *Parnassius* species (*P. acdestis*, *P. simo*, *P. orleans*, *P. nomion*, *P. apollo* and *P. glacialis*) at varying elevations from 300 to 5,000 meters a.s.l. were evaluated using genome sequencing (Table S1). The assessment results indicated that the genome sizes of the six *Parnassius* species ranged from 1.0 to 1.40 Gb (Fig. 1, Fig. S1), and species at low/median elevations possessed relatively larger genome sizes compared to those at high elevations (Fig. 1). For *P. glacialis*, the genome size was estimated to be approximately 1.33 Gb and 1.35 Gb based on the Illumina and PacBio reads, respectively (Fig. S1).

The PacBio HiFi long reads (37.54 Gb) were assembled into 1778 contigs, after which Hi-C reads (132.19 Gb) were employed to link these contigs to 169 scaffolds, resulting in a total of 29 pseudo-chromosomes (Table 1, Fig. S2). Ultimately, the chromosomal-level high-quality assembly constituted a total length of ~1.35 Gb *P. glacialis* genome and the longer scaffold N50 (49.25 Mb) (Fig. 2a). The BUSCO assessment disclosed that 95.8% of the complete single-copy genes were assembled in the *P. glacialis* genome, while 97.48% of the Illumina reads from the genome survey were found to map to 99.38% of the genome region (Table 1). Additionally, the genome collinearity revealed that 29 chromosomes of *P. glacialis* were completely mapped to 30 chromosomes of the *Papilio bianor*[24] butterfly (Fig. S3). These findings demonstrated that the *P. glacialis* genome attained a high level of assembly quality.

### Genome annotation and comparative analysis

A total of 17,080 coding genes were annotated from the 1.35 Gb *P. glacialis* genome, with 1704 genes identified as tandem repeats. Based on the NR, KEGG, GO, Pfam, and Interpro databases, 16,846 (98.60%) genes were annotated with corresponding functions. Transposable elements (TE) account for approximately 916 Mb (68%) of the *P. glacialis* genome, with long interspersed nuclear element (LINE, 527 Mb (38.96%)) being the predominant type (Fig. 2b, Supplementary Data 1). RepeatMasker analysis revealed that the TEs of three *Parnassius* species exhibited similar pattern of Kimura substitution level (%), which increased rapidly around 6 and peaked around 3 (Fig. S4a). The TE expansion in three *Parnassius* species primarily involved five types of retrotransposons: LINE/RTE, LINE/CR1, LINE/L2, LTR/Pao, and LTR/

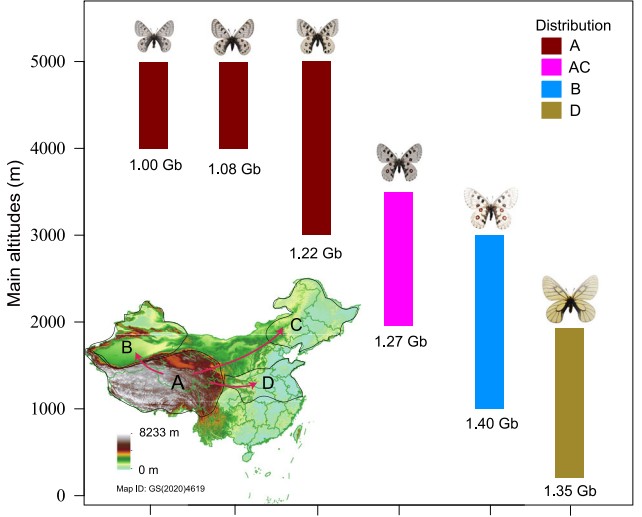

**Fig. 1 | Main altitudes and assessed genome sizes of 6 *Parnassius* species in China.** * is the selected species in this study. Map of China shows biogeographic history and general distribution of 6 *Parnassius*[18]. A: QTP; B: Xinjiang; C: Northeast China; D: East China. Genome size assessment was shown in Fig. S1. Map from the National Centre for Basic Geographic Information, official # GS(2020)4619 for public use (http://bzdt.ch.mnr.gov.cn/).

**Table 1 | Genome assembly of *P. glacialis***

| Statistics | Sequencing and assembly |
|---|---|
| Illumina reads (Gb) | 25.48 |
| PacBio reads (Gb) | 37.54 |
| Hi-C reads (Gb) | 132.19 |
| Genome size (Mb) | 1,350 |
| Contig Number | 1,778 |
| Scaffold Number | 169 |
| Chromosome Number | 29 |
| GC content (%) | 38.03 |
| Contig N50 length (Mb) | 6.54 |
| Scaffold N50 length (Mb) | 49.25 |
| BUSCO (%) | 95.8 |
| Illumina Mapping rate (%) | 97.48 |
| Illumina Mapping coverage (%) | 99.38 |

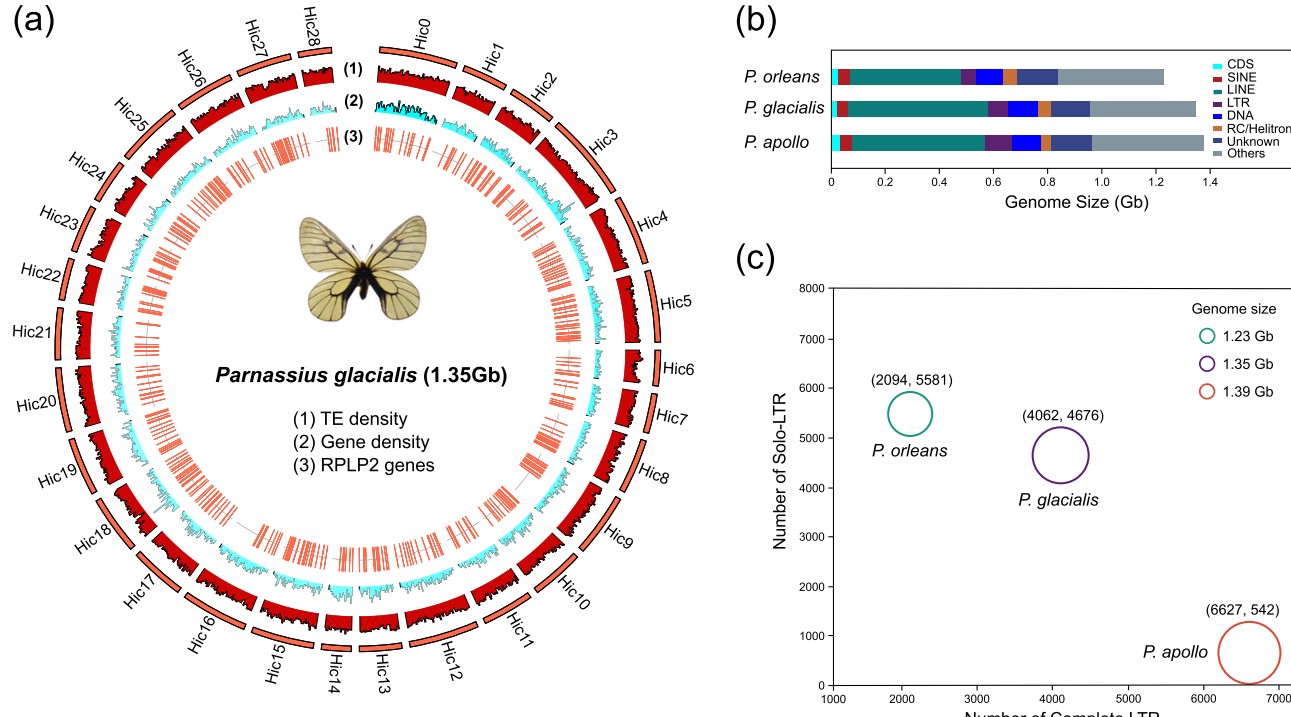

**Fig. 2 | Genome assembly and TEs comparison. a** Circos plot of genomic features and the distribution of RPLP2 genes on 29 *P. glacialis* chromosomes. TE density and gene density are counted for each 1-Mb window in *P. glacialis* genome. **b** Comparison of the genomic composition of 3 *Parnassius* butterflies. CDS stands for the coding sequences. SINE, LINE, LTR, DNA, RC/Helitron and Unknown are different types of TEs. Source data was shown in the source data file (**b**). **c** Numbers of complete LTR and solo-LTR elements in 3 *Parnassius* butterflies.

Gypsy (Supplementary Data 1). Compared to *P. orleans*, LINE/RTE, LINE/CR1, and LTR/Pao retrotransposons in *P. glacialis* increased by 36.44 Mb, 22.37 Mb, and 21.72 Mb, respectively, likely contributing to the larger genome size of *P. glacialis*.

Given that both complete LTR retrotransposons and solo-LTRs possess an intact LTR, the "80–80–80" rule[25] was employed to classify all LTR retrotransposons (Fig. S5), resulting in the identification of 303, 429, and 400 families from *P. orleans*, *P. glacialis*, and *P. apollo*, respectively. Interestingly, the number of complete LTR retrotransposons (4062) in *P. glacialis* was greater than that in *P. orleans* (2094) but smaller than that in *P. apollo* (6627) (Fig. 2c, Supplementary Data 2–7). Additionally, the number of solo-LTRs (4676) in *P. glacialis* was smaller than that in *P. orleans* (5581) but considerably larger than that in *P. apollo* (542) (Fig. 2c, Supplementary Data 2–7). These findings suggest that the unequal recombination rate[26] (solo-LTRs/complete LTRs) of LTR retrotransposons has significantly influenced the genome sizes of *Parnassius* butterflies. Moreover, the insert time analysis showed that LTR retrotransposons in all three *Parnassius* species have experienced a similar rapid increase since about 3 Ma (Fig. S4b).

## Expansion and contraction of gene families

We identified 4160 one-to-one single-copy genes among eight butterfly species and *B. mori*. Using these orthologous genes, a time tree was reconstructed with *B. mori* as the outgroup, revealing that within *Parnassius*, *P. apollo* diverged from other species around 12.33 Ma, followed by the divergence between *P. glacialis* and *P. orleans* at around 5.91 Ma (Fig. 3a). Gene family analysis demonstrated that 703, 1508, and 1303 orthologous groups expanded in *P. glacialis*, *P. apollo*, and *P. orleans*, respectively (Fig. 3a). KEGG enrichment analysis showed that the expanded orthologous groups in *P. glacialis* were significantly enriched in ribosome-associated signaling pathways (Fig. S6), mainly represented by the acidic ribosomal P protein (RPLP) gene family (Fig. 3a).

RPLP family primarily participates in protein synthesis, antioxidation, and inhibition of apoptosis[27,28], with three identified subfamilies to date (RPLP0, RPLP1, and RPLP2). Remarkably, 434 genes from the RPLP2 subfamily were identified in *P. glacialis* (Fig. 3a, Supplementary Data 8), while only one or two RPLP2 genes were found in the other seven butterfly species (Fig. 3a, Supplementary Data 9). Among these 434 RPLP2 genes widely distributed in the genome of *P. glacialis* (Fig. 2a), only one RPLP2 gene *Pglac-RPLP2* (Fig. 3b) with two introns completely aligned with the reported RPLP2 sequences of *Pa. xuthus* and *Pa. bianor* (Supplementary Data 10), while the other 433 RPLP2 genes exhibited processed pseudogene features, such as intron absence, start codon loss, or early coding termination (Fig. 3b–d, Supplementary Data 8). Most of these pseudogenes (422) lacked introns, and a few (11) contained a hypothetical intron likely generated by TE insertion (Fig. 3c). Additionally, the coverage length of these processed pseudogenes was generally less than 90% of the complete RPLP2 gene (Fig. 3d). It is worth noting that most of the RPLP2 pseudogenes (381) were found to be likely located in a gypsy family (Hic_asm_15-Gypsy-8551066-1339-5650), forming the similar chimeras with the structure of 5′-LTR, pseudogene, INTERNAL and 3′-LTR (Fig. S7, Fig. S8, Supplementary Data 11). We also found that 5′-LTRs were generally shorter than 3′-LTRs in these chimeras (Fig. S8, Supplementary Data 11).

The phylogeny of 434 RPLP2 genes in *P. glacialis* showed that the complete gene *Pglac-RPLP2* was at the basal branch of the phylogenetic tree, followed by the divergence of the other 433 processed pseudogenes (Fig. S9a). And the result of lineages-through-time (LTT) plot suggested that these processed pseudogenes underwent a dramatic expansion since about 3 Ma (Fig. S9b). Furthermore, the mean intraspecific Ka/Ks rate (0.36) of 434 RPLP2 genes in *P. glacialis* was significantly greater than interspecific value (0.08) of complete RPLP2 genes among nine lepidopteran species (Fig. 3e), which suggested that these processed pseudogenes underwent a faster evolution in *P.*

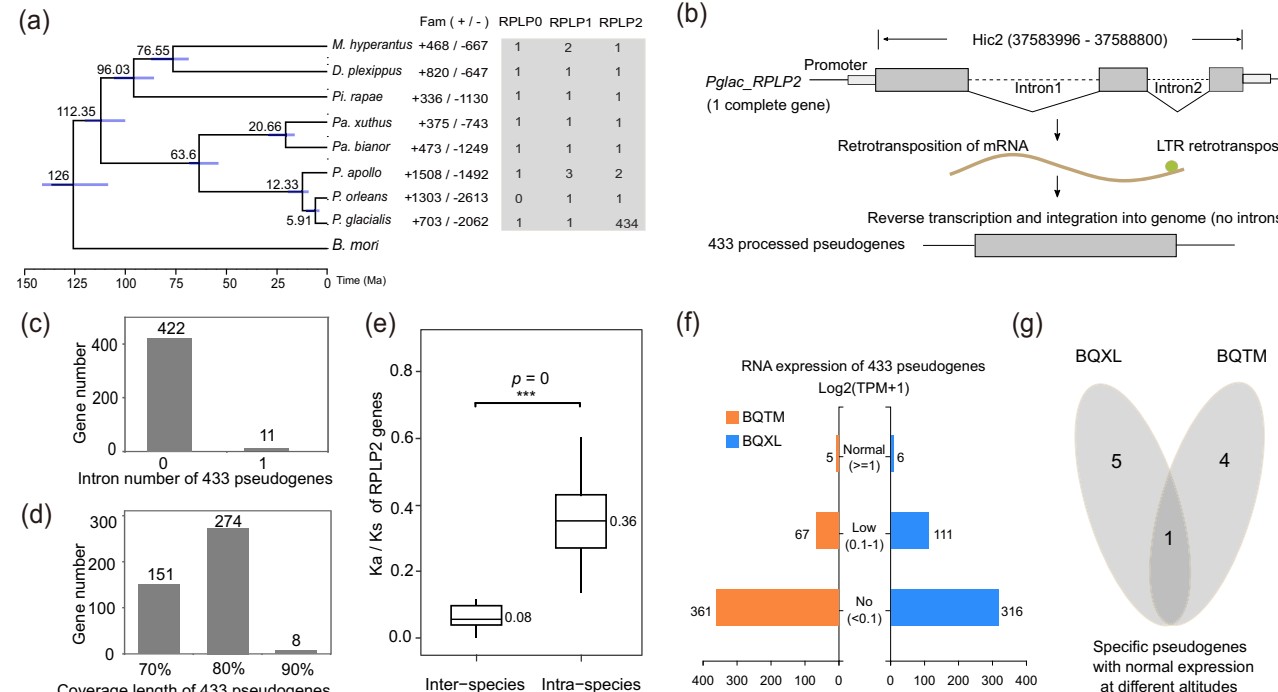

**Fig. 3 | Expansion and contraction of gene families. a** Phylogenetic tree based on single-copy genes from nine lepidopteran species. Fam (+/−) represents the number of expansion and contraction of gene families in eight butterfly species. RPLP0, RPLP1 and RPLP2 represent the number of each family in eight butterfly species. **b** 433 processed pseudogenes originated from one complete RPLP2 gene in *P. glacialis*. **c** The intron number of 433 processed pseudogenes in *P. glacialis*. **d** Coverage length of 433 processed pseudogenes in *P. glacialis*. Coverage length is the ratio between the length of translated pseudogene and the length of one complete RPLP2 protein in *P. glacialis*. **e** Comparison of Ka/Ks rates between inter-species and intra species for RPLP2 genes (*n* = 9 and 374, left to right, respectively). The Ka/Ks rates of inter-species were calculated based on the complete RPLP2

genes of *P. glacialis* (*Pglac-RPLP2*) and other eight lepidopteran species. The Ka/Ks rates of intra-species were calculated based on one complete RPLP2 gene (*Pglac-RPLP2*) and 433 processed pseudogenes in *P. glacialis* (Among them, 374 pseudogenes were calculated the valid Ka/Ks rates with *Pglac-RPLP2*). For box plots, center lines show the medians; box bounds stand for the 25th and 75th percentiles; whiskers extend 1.5 times the interquartile range from the 25th and 75th percentiles. Source data was shown in the source data file (Fig. 3e). **f** The RNA expressions of 433 processed pseudogenes at different altitudes in *P. glacialis*. BQTM and BQXL stand for the low- and high- populations. **g** Venn of specific processed pseudogenes with normal expression (log2(TPM + 1) ≥1) in the low- and high-altitude populations (BQTM and BQXL) for *P. glacialis*.

*glacialis*. Transcriptome analysis indicated that 67 and 111 processed pseudogenes were found to produce transcriptions with low expressions in low- and high-altitude populations (BQTM and BQXL) (Fig. 3f, Table S2, c 8). Notably, 5 and 6 of these pseudogenes, as well as the complete RPLP2 gene, exhibited normal expression in the BQTM and BQXL populations, respectively (Fig. 3g). Among these processed pseudogenes with normal expression, total 4 pseudogenes (*Pglac-RPLP2-1*, *Pglac-RPLP2-2*, *Pglac-RPLP2-3* and *Pglac-RPLP2-19*) showed specific expression in the low-altitude population BQTM (Fig. 3g), among them, 3 (*Pglac-RPLP2-1*, *Pglac-RPLP2-2*, *Pglac-RPLP2-3*) were clustered in the Clade1 branch of RPLP2 phylogenic tree (Fig. S9a), *Pglac-RPLP2-19* was the only one of significantly up-regulated RPLP2 gene compared to BQXL population (Fig. S9c, Supplementary Data 8). Furthermore, we identified the start codon and coding sequences of these four RPLP2 genes (Supplementary Data 12).

## Population genetic diversity and demographic history

Analysis of genome data from 41 samples (9 populations) of *P. glacialis* (Table 2, Fig. 4a, Supplementary Data 13) identified 8,425,996 high-quality SNP loci using GATK software. Based on these SNP in *P. glacialis*, the phylogenetic tree of nine populations displayed a clear altitude gradient differentiation from 1800 to 300 m (Fig. 4a, Fig. S10). Concurrently, genetic structure analysis using Admixture (Fig. 4c) showed that at k = 2, the southeast population BQTM formed one cluster while individuals from the remaining eight populations assigned to the second cluster; at k = 3, three clusters were supported, with one cluster from k = 2 containing the southeast population BQTM,

a second cluster comprising BQLS and BQKY, and a third cluster, including the other six populations mostly at higher altitudes; at k = 4, the southeast populations BQTM and BQLS each formed separate clusters, a third cluster included the northeast populations BQKY and BQTA, and the fourth cluster consisted of the remaining five populations at altitudes between 600 and 1800 m. The Admixture line graph indicated that k = 2 was the best pattern (Fig. S11), where BQTM and the other eight populations each formed a cluster, consistent with principal component analysis results (Fig. 4b). Moreover, both BQLJ and BQTT, located in central regions, appeared to have greater gene flow compared to eastern and western populations (Fig. 4c), aligning with Treemix analysis findings (Fig. S12).

Comparative analysis between high- and low-altitude populations revealed that the *Pi* values of BQXL, BQHD, and BQSN were approximately 0.00179, 0.00179, and 0.00177, respectively, all higher than the values (0.00157, 0.00159, and 0.00151, respectively) observed for BQKY, BQLS, and BQTM in *P. glacialis* (Fig. 4c). The *Pi* values of intermediate populations (BQLJ, BQTT, and BQTA) were approximately 0.00180, 0.00170, and 0.00166, respectively. These findings suggest that genetic diversity in *P. glacialis* populations at relatively low altitudes is generally lower than that in high-altitude populations. However, it is worth noting that BQLJ exhibits the highest diversity (0.00180), potentially due to secondary contact between low- and high-altitude populations, as supported by Treemix analysis results (Fig. S12).

A total of 201,680 structural variations (SVs) were identified between BQXL and BQTM populations. Of these, 140,666 (70%) SVs

**Table 2 | Population information of *P. glacialis* in this study**

| Sample ID | Species | Locality | Mean altitudes (m) | Sample number |
|---|---|---|---|---|
| BQTM | *P. glacialis* | Tianmushan, Luanchuan, Zhejiang | 300 | 10 |
| BQLS | *P. glacialis* | Laoshan, Nanjing, Jiangsu | 300 | 3 |
| BQKY | *P. glacialis* | Kunyishan, Yantai, Shangdong | 300 | 3 |
| BQTA | *P. glacialis* | Taishan, Taian, Shandong | 500 | 3 |
| BQTT | *P. glacialis* | Tiantangzhai, Jinzhai, Anhui | 600 | 3 |
| BQLJ | *P. glacialis* | Laojunshan, Luanchuan, Hena | 800 | 3 |
| BQSN | *P. glacialis* | Shennongjia, Hubei | 1800 | 3 |
| BQHD | *P. glacialis* | Huoditang, Ningshan, Shanxi | 1800 | 3 |
| BQXL | *P. glacialis* | Xiaolongshan, Maiji, Gansu | 1800 | 10 |

The geographical distribution of the listed populations is depicted in Fig. 4a.

were annotated as transposon-mediated SVs (TE-SVs), and this substantial number of TE-SVs was primarily distributed across 80,891 10-kb windows in the *P. glacialis* genome. The average $F_{ST}$ value for 10-kb windows with TE-SVs ranged from approximately 0.0553 to 0.0822 for each chromosome (Fig. 4e), which was significantly higher than those (0.0517 ~ 0.0779) without TE-SVs (No TE-SVs, $p = 0.013$) (Fig. 4e). Furthermore, the 10-kb genome windows with DNA-, LINE-, SINE-, and RC/Helitron-mediated SVs exhibited higher $F_{ST}$ values than those with LTR retrotransposons (Fig. 4f). These findings indicate that different types of transposons might have varying effects on the genetic differentiation in *P. glacialis*. Additionally, the average recombination rate ρ (0.128) of BQXL population was significantly higher than that (0.061) of BQTM population, which showed on every chromosome in *P. glacialis* (Fig. S13a). The regions of No TE-SVs were found to have higher recombination rates (0.081 and 0.148) than TE-SVs' (0.049 and 0.118) in both BQTM and BQXL populations (Fig. S13b). Furthermore, the recombination rate showed negative correlation ($P < 0.01$) with $F_{ST}$ value in *P. glacialis* (Fig. S13c).

PSMC results (pairwise sequentially Markovian coalescence model) revealed the historical population dynamics of *P. glacialis* at different altitudes (Fig. 4d). Approximately 200 to 80 thousand years ago (ka), the historical population sizes of *P. glacialis* reached their peak values. The population size of BQXL increased after the Last Glacial Maximum (LGM, 26.5–19 ka)[29], while the BQTM exhibited a persistent decline. Overall, the population size of relatively low-altitude populations was generally smaller than that of high-altitude populations after the LGM.

**Selective sweeps for the BQTM population**

Based on the SNPs obtained for the BQXL and BQTM populations, a total of 64,833 20-kb windows, each containing a minimum of 10 SNPs, were filtered to calculate the values of *Pi* ratios, $F_{ST}$, XPEHH and Tajima's D (Supplementary Data 14). For the population BQTM, we identified 646, 645, 644 and 647 candidate genes with selective signature by the bottom 1% of *Pi* ratios, the top 1% of $F_{ST}$ values, the top 1% of XPEHH values and the bottom 1% of Tajima's D values, respectively (Fig. 5a). Among them, 454 of these candidate genes were supported by at least two methods (Fig. 5a) and significantly enriched into several pathways ($P < 0.05$) (Fig. 5b, Supplementary Data 15). Among these genes (Supplementary Data 15), we found 9 genes (8 *Anpep* and 1 *LAP3*) involved in the glutathione metabolism pathway[30], 6 genes (*Ds, Ed, Lgl, Mer, PatJ* and *Sdt*) located in the hippo signaling pathway-fly pathway[31], and 6 genes (*PSH, Toll, P38, Dl/Dif* and 2 *PGRP*) located in the toll and Imd signaling pathway[32] (Fig. 5b). Notably, we found that some genes with selective signature belonged to tandem gene duplications (TGDs), such as *Anpep* and *PGRP*. Take the glutathione metabolism pathway as an example, 8 *Anpep* TGDs on chromosome 25 showed obviously selective signatures with lower *Pi* ratios, higher $F_{ST}$ values, higher XPEHH values and lower Tajima's D values than that of the regions on

both sides (Fig. 5c, e). Furthermore, we compared the number of TE-SVs located in the genes of *P. glacialis* (Fig. 5d). No difference in number of TE-SVs (about 2) was found in the regions of upstream (Up-3 kb) and downstream (Down-3 kb) for all genes with or without selective signature (Fig. 5d). However, 454 genes with selective signature harbor significantly more TE-SVs (6.5) than that (5) of others without signature (Fig. 5d). These results indicated that selective signature was associated with the number of TE-SVs located in these genes.

## Discussion

Prior research has emphasized the significant impact of transposable elements (TEs) on genome evolution[33–36]. TE activity can alter genome size and structure, profoundly influencing the evolutionary trajectory of host organisms[37,38]. In the current study, three *Parnassius* species were found to exhibit similar TE expansion patterns at the distribution of Kimura substitution level, as well as similar rapid increasing of LTR retrotransposons since about 3 Ma during the early Quaternary Ice Age (QIA) (Fig. S4b). Previous studies have shown that environmental stress is a major factor influencing transposon activity[39,40]. During the cyclonic glacial and interglacial periods of QIA, these *Parnassius* species (like *P. glacialis* or *P. apollo*, etc.) spreading out of the QTP had to face more pressure of climate change due to the loss of their nearby natural refuge of the vertical altitude gradient on the QTP. Therefore, the increased environmental stress experienced by *P. glacialis* and *P. apollo* might explain the increased TE content in their genomes. Furthermore, we discovered that only a few transposon types, including LINE/RTE, LINE/CR1, and LTR/Pao, etc (Supplementary Data 1), primarily contribute to genome size variation in these *Parnassius* species, which is consistent with previous studies[41]. Additionally, earlier research indicated that insertion and deletion of LTR retrotransposons were relatively balanced in the genome, and an increasing unequal recombination rate (solo-LTRs/complete LTRs) could affect genome size reduction[26]. In comparison with high-altitude *P. orleans* species, the lower unequal recombination rate of LTR retrotransposons in *P. glacialis* and *P. apollo* at relatively low altitudes (Fig. 2c) could lead to increased TE content, potentially contributing to the formation of larger genome sizes in these *Parnassius* species spreading out of the QTP (Fig. 1).

Our findings reveal that numerous RPLP2 processed pseudogenes were identified to come from one complete functional RPLP2 gene in the *P. glacialis* genome (Fig. 3a–d, Fig. S9a). The phylogeny tree and LTT plot (Fig. S9b) showed that most of these pseudogenes presented rapid divergence since about 3 Ma during the early Quaternary Ice Age, which was generally consistent with the timing of TEs activity in *P. glacialis* (Fig. S4b). Earlier studies demonstrated that processed pseudogenes typically result from retrotransposition mediated by LINE transposons[42] or LTR retrotransposons[43]. Considering that most of the RPLP2 pseudogenes were likely located in the region of LTR retrotransposons (Fig. S8, Supplementary Data 11), we hypothesize

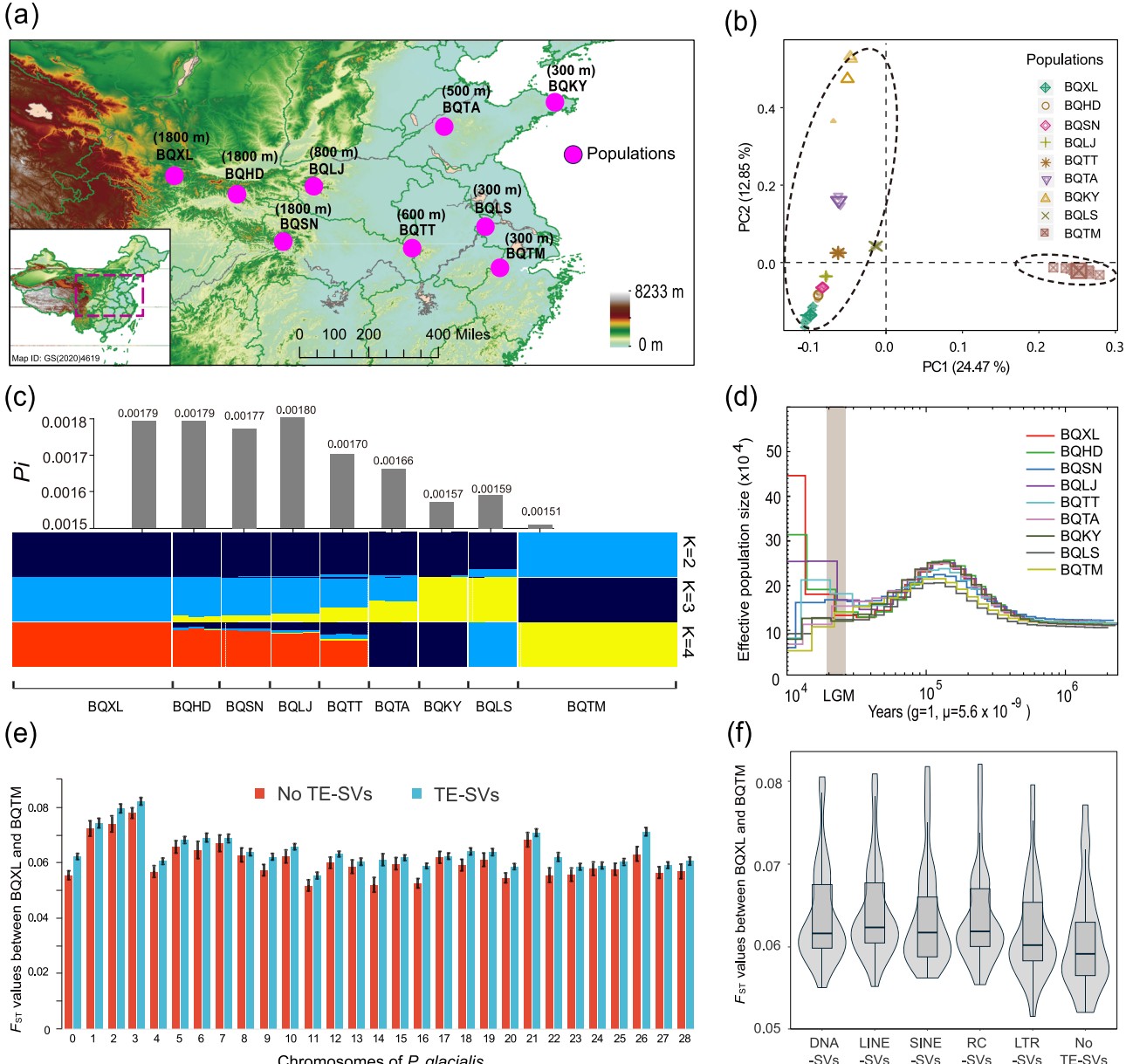

**Fig. 4 | Historical dynamics and genetic diversity of different altitude populations in *P. glacialis*. a** Geographic locations and main altitudes of sampled populations (Table 2). Map from the National Centre for Basic Geographic Information, official # GS(2020)4619 for public use (http://bzdt.ch.mnr.gov.cn/). **b** Principal component analysis for nine *P. glacialis* populations. **c** Nucleotide diversity (*Pi*) and genetic structure of nine *P. glacialis* populations. **d** Historical population dynamics for nine *P. glacialis* populations. **e** Comparison of average $F_{ST}$ values (BQXL and BQTM) with and without transposon-mediated SVs across 29 chromosomes in *P. glacialis*. TE-SVs and No TE-SVs represent the 10-kb genome windows with and without transposon-mediated SVs in *P. glacialis*, respectively. The error line of bar plot represents the 95% confidence interval. Source data was shown in the source data file (**e**). **f** Impact of different transposons on genetic differentiation in *P. glacialis*. DNA-, LINE-, SINE-, RC/Helitron-, and LTR- SVs represent various types of transposon-mediated SVs in *P. glacialis*. Average $F_{ST}$ values of 10 kb-windows with different transposons for each chromosome: $n = 29, 29, 29, 29, 29$ and $29$, left to right, respectively. For box plots, center lines show the medians; box bounds stand for the 25th and 75th percentiles; whiskers extend 1.5 times the interquartile range from the 25th and 75th percentiles. Source data was shown in the source data file (**f**).

that the RPLP2 gene family generates numerous processed pseudogenes through these retrotransposons (Fig. 3a, b). Although most pseudogenes are thought to be nonfunctional, a few transcribed pseudogenes have been reported to obtain the defined functions through rapid evolution[44]. Our results reveal that most RPLP2 processed pseudogenes in *P. glacialis* harbor faster evolutionary rate (Ka/Ks) than that of complete RPLP2 genes among eight butterflies (Fig. 3e), and a few of them display normal expressions as the functional genes in *P. glacialis* (Fig. 3f, g). It is particularly interesting to note that there are several specific expressed pseudogenes (*Pglac-*

*RPLP2-1*, *Pglac-RPLP2-2*, *Pglac-RPLP2-3* and *Pglac-RPLP2-19*) with start codons and one of them (*Pglac-RPLP2-19*) was significantly up-regulated in the low altitude population BQTM (Fig. S9c, Supplementary Data 8, Supplementary Data 12). Given that the RPLP2 gene is known to promote protein synthesis and is linked to antioxidation and inhibition of cellular senescence[27,28], we hypothesize that these RPLP2 processed pseudogenes have undergone rapid evolution to obtain the ability of functional genes, possibly as a response of *P. glacialis* to the low-altitude environments outside the QTP, such as physical development, oxidative damage and pathogenic microorganism invasions.

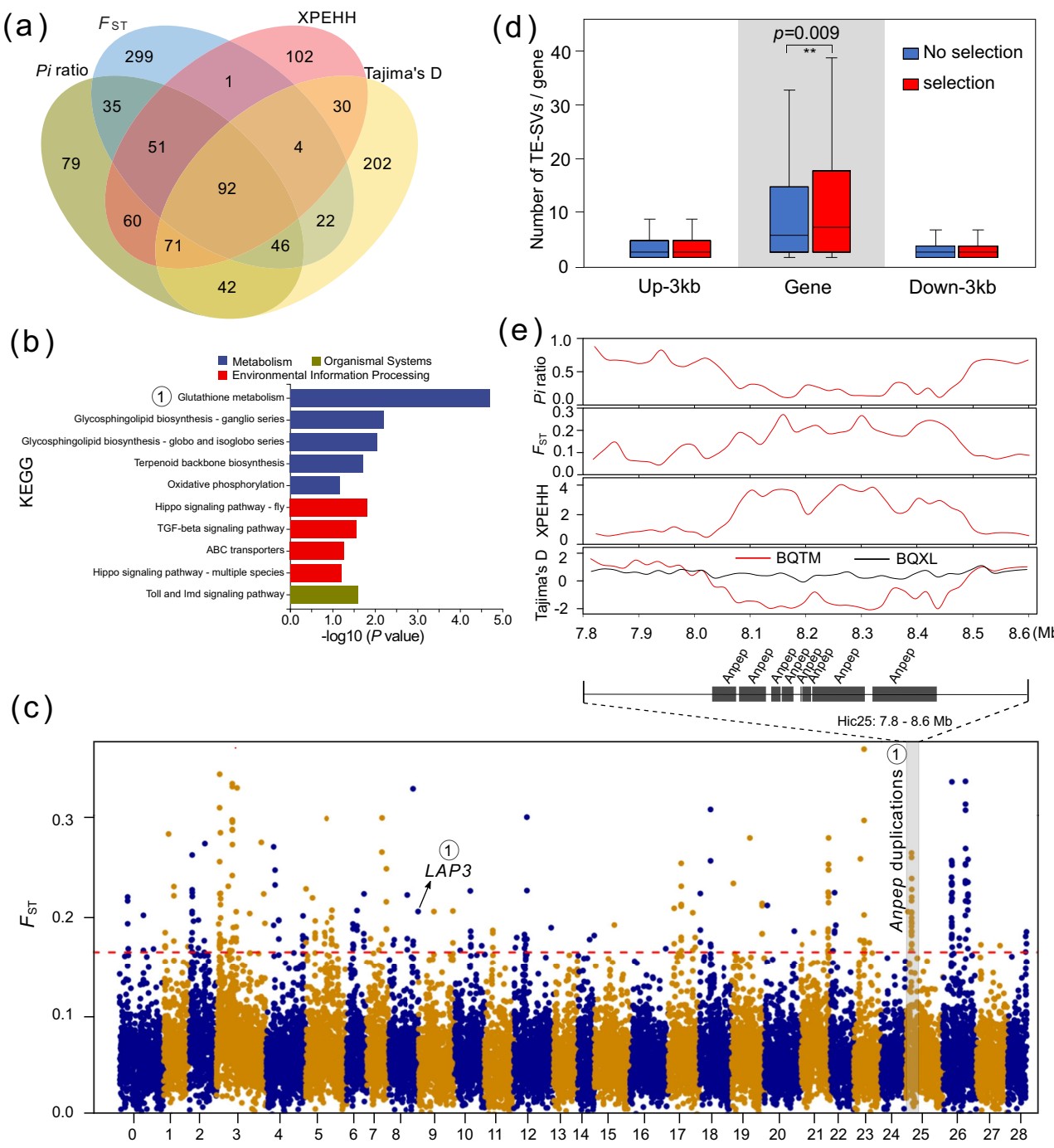

**Fig. 5 | Selective signature in the low-altitude population BQTM. a** Venn plot of candidate genes identified by four methods of *Pi* ratios, $F_{ST}$, XPEHH and Tajima's *D*. **b** KEGG enrichment of candidate genes supported by at least two methods. The exact *P* values were shown in Supplementary Data 2. **c** Manhattan plot of $F_{ST}$ values between BQXL and BQTM. The red dotted line (0.1711) represents the top 1% of $F_{ST}$ values. The symbol ① represents the glutathione metabolism pathway. **d** Comparison of TE-SVs' number for genes with and without selective signature ($n$ = 6851, 186, 9803, 307, 6523 and 165, respectively). Up-3 kb, Gene and Down-3 kb represent the TE-SVs' location in the regions of upstream 3 kb, gene and down 3 kb, respectively. For box plots, center lines show the medians; box bounds stand for the 25th and 75th percentiles; whiskers extend 1.5 times the interquartile range from the 25th and 75th percentiles. Source data was shown in the source data file (Fig. 5d). **e** Selective signature of *Anpep* duplications on chromosome 25. BQTM and BQXL represent the low- and high- populations in *P. glacialis*.

Earlier studies have shown that populations at different altitudes exhibit distinct biodiversity and population dynamics due to alpine geological isolation and local adaptation effects[45]. The population size of *P. glacialis* at relatively high altitudes was similar to that at low altitudes before the LGM, with an increase in population size for high altitude populations (like BQXL) following the LGM (Fig. 4d). This may

be attributed to the Qilian Mountains at the northern margin of the QTP becoming a natural refuge after the LGM and the central China population's spread to this refuge, as supported by Treemix analysis results (Fig. S12). A similar population dynamic has been previously described for the alpine butterfly species *Lycaena tityrus*[46]. However, unlike the BQXL population, under the trend of global warming after

the LGM, the *P. glacialis* populations (like BQTM) at the low altitudes likely experienced greater selection pressures, such as increasing temperature, humidity, and other related factors, resulting in a decreased population size (Fig. 4d). This finding suggests that *P. glacialis* dispersed into low-altitude areas during the cold period of late QIA based on ancestral geographic reconstruction[18], and subsequently underwent a genetic bottleneck after the LGM, with only a small number of adapted individuals surviving in these regions (Fig. 4c, d). Concurrently, genome regions with TE-SVs were found to harbor lower recombination rate and higher $F_{ST}$ values than those without (Fig. 4e, f, Fig. S13b), indicating that TE activity probably tended to increase the genetic differentiation between high- and low-altitude *P. glacialis* populations. Moreover, our results showed that lower recombination could also lead to higher differentiation and more TE-SVs (Fig. S13b, c), which is consistent with previous studies[47]. Given the coevolution between TE and recombination in previous studies[48], we suggest that they both contribute to influence the genetic differentiation in *P. glacialis*. Additionally, a higher value of recombination rate $\rho$ (=$4N_e*r$) in BQXL than BQTM (Fig. S13a) could either be caused by higher crossover recombination rate (*r*), or larger effective population size ($N_e$). Given that *r* usually evolves quite slowly, it seems more likely that the difference in $\rho$ is caused by a difference in $N_e$, which is also supported by the PSMC result (Fig. 4d).

Furthermore, a series of genes with selective signature were identified for the low-altitude BQTM population (Fig. 5a), enriching in several pathways associated to the function of antioxidant, development and immune (Fig. 5b, Supplementary Data 15). For example, the glutathione metabolism pathway was reported to play an important role in the cellular protection against oxidative stress in *Drosophila*, such as high oxygen[30], high temperature[49], heavy metal pollution[50], ionizing radiation[49], and chemical toxins[51]. Considering the conservative function of glutathione metabolism pathway in insects, we speculate that its genes with selective signature probably reflected the response of *P. glacialis* to the low-altitude environments, such as enriched oxygen content, warmer climate and more toxic honey source plants. Another example is the hippo signaling pathway (Fig. 5b, Supplementary Data 15), initially discovered in *Drosophila*, has been recognized as a conserved signaling pathway that controls organ size during development by restricting cell growth and proliferation and by promoting apoptosis[31]. In our study, we speculate that these genes with selective signature involved in the hippo signaling pathway might be related to the body size variation for *P. glacialis* in low-altitude regions. Otherwise, the toll and Imd signaling pathway (Fig. 5b, Supplementary Data 15) was reported to play a role in regulating the immune system in *Drosophila melanogaster*[32], we speculate that its genes with selective signature in *P. glacialis* of low altitudes is likely related to enhancing the immune defense against increased pathogenic agents in humid and warm climates in these areas. Moreover, the association of these selections with TE-SVs (Fig. 5d) gives us a possible explanation that TE-SVs might provide more opportunities for selective evolution of *P. glacialis* to meet challenges of new habitats.

In conclusion, our investigation of the chromosome-level *P. glacialis* genome and resequenced genomic data offers insights into genome evolution and local adaptation of this alpine butterfly species. Our findings suggest that TEs may have a crucial role in genome evolution, including genome size variation, processed pseudogene expansion, and population genetic differentiation in *P. glacialis*. Additionally, the transposon-mediated genetic differentiation probably provided an opportunity for selective sweeps and potential local adaptation of *P. glacialis*. These results not only enhance our comprehension of how *P. glacialis* has dispersed to southern regions of China but also supply a robust molecular foundation for future research on the evolution and adaptation of other alpine organisms originating from the QTP.

## Methods

### Sample collection and sequencing

Two 5th instar larvae of *P. glacialis* were collected from an altitude of 300 m a.s.l. in Laoshan, Nanjing, China. One larva was starved for 48 h and then rapidly frozen in liquid nitrogen until it was used for genome survey and the de novo genome sequencing. The other larva was used for Hi-C sequencing (Table S1). We collected 25 adult individuals, and combined them with the reported genome sequencing of 16 *P. glacialis* samples in the NCBI SRA database[52], resulting in 41 samples from 9 populations spanning from western to central and eastern China at altitudes ranging from 300 to 1800 m a.s.l. (Table 2, Supplementary Data 13). All new samples were preserved in 95% ethanol until used for genome resequencing. Our research complies with government regulations on animal protection while sampling all specimens in this study. No permits were required for collection of these species.

The total genomic DNA of *P. glacialis* was extracted from the thorax muscle of the insect samples using the QIAamp DNA Purification Kit (Qiagen). Whole-genome sequencing was performed using SMRT sequencing technology (Pacific Biosciences) and Illumina sequencing technology (Illumina, San Diego, CA, USA). Long-read libraries with a fragment size of 20 kb were constructed using the SMRTbell Template Prep Kit (Pacific Biosciences), while 150 bp paired-end libraries with an insert size of 350 bp were constructed using the TruSeq Nano DNA Library Prep Kit (Illumina). The 20 kb and 150 bp paired-end libraries were sequenced using the PacBio HiFi and Illumina HiSeq X Ten instruments, respectively (Table S1, Supplementary Data 13).

The raw data was filtered to remove reads with adaptor, low-quality reads and duplicated reads using FastQC (https://github.com/s-andrews/FastQC/). The QC procedures were as follows: (a) removal of reads with ≥10% unidentified nucleotides (N), (b) removal of reads with >20% of bases with a Phred quality <5, (c) removal of reads with >10 nucleotides aligned to the adapter, allowing ≤10% mismatches, and (d) removal of putative PCR duplicates generated by PCR amplification during the library construction process.

### Genome survey and assembly

To assess genome size, we downloaded the genome sequencing data (Illumina sequencer) for five *Parnassius* species (*P. acdestis*, *P. simo*, *P. orleans*, *P. apollo* and *P. nomion*) from NCBI SRA database (Table S1). For *P. glacialis*, we used the sequencing data from Illumina and PacBio sequencers for assessment, respectively (Table S1). Jellyfish 2.2.10[53] and GCE software v1.0.0[54] were used to estimate the genome sizes of these six *Parnassius* species based on their respective 17 k-mer frequency distributions (Fig. S1). The PacBio reads were assembled into contigs of the *P. glacialis* genome using Hifiasm v0.13[55] with the "-k 51" option, and Clean Hi-C reads were mapped to draft genomes using BWA v0.7.12[56]. The chromosome-level genome was clustered using the contigs according to the unique, high-quality paired-end reads of Hi-C in the ALLHiC software operated at default parameters[57].

In order to evaluate the quality of the prepared genome assemblies, we aligned the Illumina sequencing reads of the genome survey to the assembled *P. glacialis* genome using BWA v0.7.12[56]. The integrity of the genome assembly was also assessed using BUSCO v5.0.0 (http://busco.ezlab.org/) based on the insecta_odb10 datasets[58]. According to the gene annotations of *P. glacialis* and *Pa. bianor*[24], the chromosome collinearity was constructed using the tools JCVI v1.3.4 (Fig. S3)[59].

### Genome annotation and TE analysis

To construct the repeat library of three *Parnassius* species, the genomic sequences of *P. apollo* (GCA_907164705.1)[16] and *P. orleans*[15] were downloaded from the NCBI genome database (https://www.ncbi.nlm.nih.gov/genome) and ScienceDB (https://www.scidb.cn/en/cstr/31253.11.sciencedb.o00023.00001). The software RepeatModeler v2.0.1[60], LTR_Finder v1.05[61], LTRharverst[62], and RepeatScout v1.0.5[63] were used

with default parameters. The repeat sequences were obtained by merging the RepBase-20181026 databases (https://www.girinst.org/repbase/). The total repetitive sequences of three *Parnassius* butterfly species were predicted using RepeatMasker v4.0.6 (-nolow)[64]. The Kimura substitution level was calculated based on the downstream scripts of RepeatMasker using default parameters. The annotation of complete LTR and solo-LTR retrotransposons was illustrated stepwise in Fig. S5. The 5′- and 3′-LTRs sequences for each complete LTR retrotransposon were aligned by MAFFT v7[65] to calculate the pairwise distance (k) using the Kimura model by EMBOSS v6.6[66]. The insertion time (*T*) of complete LTR retrotransposons was estimated using the formula $T = k/2r$ based on a substitution rate of $r = 5.6 \times 10^{-9}$ /per site/per year, as described in previous studies[18].

Gene annotation was performed using three methods: ab initio prediction, homology alignment, and RNA-seq support. Tools of Geneid v1.4[67], Genescan v1.0[68], GlimmerHMM v3.04[69], SNAP v2013[70], and Augustus v2.4[71] were used for ab initio gene prediction. The homolog method was carried out using the software GeMoMa v1.3.1[72] with default parameters. Transcriptome sequences of 6 adult individuals (muscle isolated from thorax) for two *P. glacialis* populations (BQXL and BQTM) were downloaded from the NCBI SRA database (Table S2)[73]. Transcript assembly and expression analysis based on the reference genome were performed using Hisat v2.0.4 (-max-intronlen 20000, -min-intronlen 20)[74], Stringtie v1.2.3 (with default parameters)[75]. Gene prediction was performed using the software Transdecoder v2.0 (http://transdecoder.github.io) and Genemarks-t v5.1[76] with default parameters. The EvidenceModeler software[77] was used to generate comprehensive non-redundant genes by integrating all genes predicted by the three methods. Functional annotations were obtained by searching the databases of NCBI-NR, Gene Ontology (GO)[78], Kyoto Encyclopedia of Genes and Genomes (KEGG)[79], SwissProt[80], and Pfam[81].

## Phylogeny and gene family analysis

The genomic sequences of *Maniola hyperantus* (GCA_902806685.1), *Danaus plexippus* (GCA_018135715.1)[82], *Pieris rapae* (GCA_905147795.1), *Papilio xuthus* (GCA_001298345.1)[17], and *Bombyx mori* (GCA_014905235.2)[83] were procured from the NCBI genome database (https://www.ncbi.nlm.nih.gov/genome). Genome sequences of *Papilio bianor*[24] were acquired from public website (https://ftp.cngb.org/pub/gigadb/pub/10.5524/100001_101000/100653) of GigaDB. Protein sequences from eight butterfly species (*M. hyperantus*, *D. plexippus*, *Pi. rapae*, *Pa. xuthus*, *Pa. bianor*, *P. apollo*, *P. orleans*, and *P. glacialis*) and *Bombyx mori* were analyzed to obtain one-to-one orthologous genes using Orthofinder[84]. The sequences were aligned in MAFFT v7[65] and processed in trimAl[85]. Subsequently, the phylogenetic relationship among nine representative lepidopteran species was constructed with RAxML v8.2.10[86] employing 100 bootstrap replicates under the GTRGAMMA model. Considering that there were fewer fossils of butterflies, we selected two calibration points based on previous studies[5,87]: (1) the crown of *Parnassius* (10.5–16.6 Ma)[5]; (2) the crown of Papilionoidea (110.3–86.9 Ma)[87]. The time tree of nine lepidopteran species was reconstructed using the MCMCtree program in PAML v4[88]. Based on the Orthofinder analysis results, gene family contraction and expansion were examined using Café (*P* < 0.01)[89] and ultimately visualized with Figtree v1.4.3[90].

The initial results of Café showed a significant expanded group containing 17 RPLP genes in *P. glacialis*, while the other species had only one or two. At the same time, we found some RPLP genes without introns (like pseudogene) in *P. glacialis*. In order to identify all the pseudogene sequences of expanded gene family RPLP, the annotated RPLP sequences (including RPLP0, RPLP1 and RPLP2) of *Pa. xuthus* and *Pa. bianor* were retrieved from the NCBI database. These protein sequences were then aligned to the genomes of nine lepidopteran species using Tblastn of Blast v2.9.0[91], and candidate sequences were

acquired by extending 5 kb on both sides. Gene structure prediction was performed using Exonerate v2.2.0[92] and Genewise v2.4.1[70]. The predicted RPLP protein sequences were filtered based on the Pfam domain PF00428 and coverage length (≥70%) with the reported sequences of *Pa. xuthus* and *Pa. bianor*. Based on the conserved coding sequences, the RPLP2 genes were divided into two types: complete RPLP2 genes (coverage length ≥95%, with promoter region and intron) and fragmented RPLP2 pseudogenes (coverage length <95%, without promoter region or intron).

The non-synonymous rate (Ka), synonymous rate (Ks), and Ka/Ks values of RPLP2 genes from nine lepidopteran species were calculated using KaKs_Calculator v2.0[93]. The maximum-likelihood phylogenetic tree of RPLP2 genes in *P. glacialis* was reconstructed with RaxML v8.2.10[86] by using *P. orleans* and *P. apollo* as the outgroup. On the basis of the *Parnassius*' time priors from the phylogeny tree of nine lepidopteran species in this study, the divergence time of RPLP2 genes was estimated using BEAST v1.83[94]. The MCMC chain was run for 10 million generations to achieve convergence and was sampled at every 1000 generations. The RNA expression of RPLP2 genes in high- and low-altitude populations (BQXL and BQTM) were obtained from the gene annotation results of Hisat v2.0.4[74] and Stringtie v1.2.3[75]. These processed pseudogenes were divided into three groups based on the RNA expressions (Fig. 3f): (1) Normal expression (log2(TPM + 1) ≥ 1); (2) Low expression (log2(TPM + 1) ≥ 0.1 and log2(TPM + 1) < 1); (3) No expression (log2(TPM + 1) < 0.1). Additionally, edgeR (*P* < 0.05) was used to identify the RPLP2 genes with differential expression[95].

## Identified the association of RPLP2 pseudogene with LTR retrotransposon

Gene duplication mediated by LTR retrotransposon has the hallmark repeats of LTR retrotransposon in their flanking regions[43]. Firstly, the 8 kb flanking sequences of each RPLP2 gene were aligned to check for this repeat by BLAT[96]. Secondly, the host LTR retrotransposons inserted by RPLP2 pseudogene were identified stepwise in Fig. S7. In the analysis, two major gypsy families (Hic_asm_15-Gypsy-8551066-1339-5650 and Hic_asm_0-Gypsy-28602735-1211-5694) with the open reading frame (ORF) of group-specific antigen (GAG) and polymerase (POL) were found to associate with the RPLP2 pseudogenes in *P. glacialis* (Supplementary Data 16). Considering the high similarity (97.8%) of reverse transcriptase domain (RT)[97] between the two families (Supplementary Data 17), we finally used the longer one (Hic_asm_15-Gypsy-8551066-1339-5650) to identify the structure of LTR-mediated RPLP2 pseudogenes by RepeatMasker v4.0.6 (-nolow -cutoff 600)[64] (Figs. S7, S8, Supplementary Data 11).

## Population genetics and demographic history analyses

We obtained a total of ~489 Gb clean reads from 25 sequencing individuals at ~19.5 Gb (14X) per individual (Supplementary Data 13). Combined with the reported genome sequencing of 16 *P. glacialis* samples[52], these population sequences of *P. glacialis* were aligned to reference genomes using BWA v0.7.12[56]. Following variant calling with SAMtools v1.3.1[98], the demographic histories of *P. glacialis* were ascertained utilizing PSMC v0.6.5 software[99] with a mutation rate of $5.6 \times 10^{-9}$ and one generation per year[18]. SNPs were extracted and filtered employing GATK v4.0 (QD < 2.0||FS > 60.0||MQ < 40.0||SOR > 3.0 || MQRankSum←12.5 || ReXHPosRankSum←8.0)[100]. After further SNP filtering using Plink v1.9[101] by the options (window size 10 kb, step size 10 kb, and threshold 0.5) of linkage disequilibrium (LD), the genetic structure and phylogenetic tree of nine *P. glacialis* populations were constructed with Admixture v1.3.0[102] and IQtree (MFP + ASC)[103]. Principal component analysis was then conducted using Plink v1.9[101], and gene flow was examined with Treemix v1.13[104]. Subsequently, Vcftools v0.1.17[105] was used to calculate the population nucleotide diversity (*Pi*) values.

In order to identify the signatures of selective sweep, four methods (*Pi* ratio, $F_{ST}$, XPEHH and Tajima's D) were used for the high- and

low-altitude populations (BQXL and BQTM) as previous studies[106]. Considering that the average gene length of *P. glacialis* is about 21 kb, the window is set to 20 kb for this analysis. PopgenWindows (https://github.com/simonhmartin/genomics_general/) was employed to calculate the *Pi* ratios (BQTM/BQXL) and $F_{ST}$ values for each 20-kb non-overlapping window containing at least 10 SNPs. For the *Pi* ratios (BQTM/BQXL), top 1% of them suggests selection in population BQXL, whereas bottom 1% of them suggests selection in population BQTM. XPEHH statistics were calculated using selscan[107], and then the average XPEHH score was estimated for each 20-kb non-overlapping window. For these XPEHH scores between BQTM and BQXL, a positive value (top 1% of all) suggests selection in population BQTM, whereas a negative value (bottom 1% of all) suggests selection in population BQXL. Tajima's D values were calculated for each 20-kb non-overlapping window in populations BQTM and BQXL using Vcftools v0.1.17[105]. Based on the bottom 1% of *Pi* ratios (BQTM/BQXL), top 1% of $F_{ST}$ values, top 1% of XPEHH values and bottom 1% of Tajima's D values (BQTM), the candidate regions of selective sweep supported by at least two methods were identified for the low-altitude population (BQTM). Subsequently, these regions extended 20 kb at both sides were assigned to corresponding genes using Bedtools v2.26.0[108].

### Detection of transposon-mediated structural variations

Structural variations (SVs) of the 20 samples from BQXL and BQTM populations were identified using two independent methods: (i) manta[109], and (ii) the smoove pipeline (https://github.com/brentp/smoove) which is based on lumpy[110]. All SV callers were executed using default parameters. VCF outputs were formatted and filtered (SVs >=50 bp and SVs <=100 kb) utilizing the reported scripts[111]. The two VCFs were combined using jasmine[112] with default parameters, retaining SVs supported by both approaches. If at least 90% of the SV locus sequence matched a TE, it was annotated as TE-SV, >0 to 90% as uncertain, and 0 as No TE-SV. To analyze the relationship between TE-SV and recombination rate ($\rho$), we calculated the recombination rate ($\rho = 4N_e * r$) of high- and low-altitude populations (BQXL and BQTM) by the tool LDhelmet[113] as previous studies[114,115]. Among them, $N_e$ is the effective population size and *r* is the crossover recombination rate per generation per bp[115].

### Reporting summary

Further information on research design is available in the Nature Portfolio Reporting Summary linked to this article.

## Data availability

The genome sequencing and assembled data in this study have been deposited in the BioProject (PRJNA893814) of GenBank database. The Supplementary Figures and Tables in this study are provided in the PDF file of Supplementary Information. The Supplementary files in this study are provided in the files of Supplementary data 1-17. The Source data of Figs. 2b, 3e, 4e, 4f, 5d, S4a, S4b, S6, S13a and S13b are provided as a Source Data file. Source data are provided with this paper.

## Code availability

The script used in this work is available at Github.

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

## Acknowledgements

We thank Prof. Huabin Zhao (Wuhan University, China) and Xuhua Xia (Ottawa University, Canada) for his kind suggestions about the manuscript's writing and Dr. Luyan Li and Miss Zhen Zhao (Nanjing Institute of Geology and Paleontology, CAS, China) for their help in field specimen collection. This work received financial support from the National Science Foundation of China (Grants No. 41972029 to J.H. and No. 31960142 to Y.Z.), the CAS Strategic Priority Research Program (Grant No. XDB26010204 to Q.Y.), the State Key Laboratory of Paleobiology and Stratigraphy (Nanjing Institute of Geology and Paleontology, CAS) (Grant No. Y626040108 to Q.Y.) and the National Science Foundation of Anhui (Grant No. KJ2021A0100 to C.S.).

## Author contributions

J.H. and Y.Z. planned the study. Y.Z., C.S. and B.H. performed the laboratory work. Y.Z. and B.H performed the computational analyses. Y.Z. and C.S. drafted the manuscript. R.N., J.H., Y.W., J.S., J.M., and Q.Y. helped to revise the manuscript. J.H. and Q.Y. provided the major funding support. All authors have read and agreed to the published version of the paper.

## Competing interests

The authors declare no competing interests.
