## [Peer Review File · Nature Communications]

Dispersal from the Qinghai-Tibet Plateau by a high-altitude butterfly is associated with rapid expansion and reorganization of its genomeREVIEWER COMMENTS

Reviewer #1 (Remarks to the Author):

Youjie Zhao and co-authors investigated genome evolution and population history of *Parnassius glacialis*, a butterfly with an exceptionally large genome that is thought to have expanded its range outward from the Qinghai-Tibet Plateau. They compare genome size and content between *Parnassius* species, describe population structure and differentiation between various *P. glacialis* populations at different altitudes, and discuss evidence that TEs play a role in local adaptation.

This is an interesting study system for investigating genome evolution and local adaptation. The authors have generated a powerful dataset, and provided a comprehensive description of the genome and population genomic patterns, which would make this work of interest to others interested in genome size evolution and local adaptation. However, I believe that some of the claims in this paper, including the claim in the title that local adaptation is “driven extensively by transposable elements”, have not been adequately proved, because the authors do not consider and test alternative explanations for the observations. Below I describe in detail what I mean by this, and suggest ways the authors could improve the support for their claims. I also highlight several areas where the manuscript provides insufficient detail to describe analyses or interpretations. I believe all of these concerns can be addressed with some additional analyses and modifications to the text.

1. local adaptation

The authors find that SV regions overlapping TEs, have higher differentiation (F_{st}) between altitudes on average. They state:

“These findings indicate that different types of transposons might have varying effects on accelerating genetic differentiation in *P. glacialis*.” (Section 2.4)

“TE activity may drive genetic differentiation between high- and low-altitude *P. glacialis* populations.” (Section 3)

This hypothesis is plausible, but it is not the only possible explanation for the observed results. It is well known that regions of lower recombination rate tend to have higher differentiation due to lower effective population size (e.g. Burri et al. *Evolution Letters* 2017 <https://doi.org/10.1002/evl3.14>). Therefore an alternative interpretation of these results is that TEs might be more common in regions of low recombination rate. In other words, TEs might not be causal of the increased differentiation, but rather a consequence of lower recombination rate. The authors need to discuss this alternative interpretation of the results. In order to claim that TEs drive local adaptation, I would further recommend testing the alternative interpretation by examining estimated recombination rates. For example see how this was tested by Montejo-Kovacevich et al. 2022 *Nat. Comm.* <https://doi.org/10.1038/s41467-022-32316-x> (Figure 3c).

2. Transposon-mediated SVs

Section 5.6: “Regions where SVs overlapped with TEs were annotated as transposon-mediated SVs”

Please provide further detail here. Was it necessary that the SV was equivalent in size to the TE? If not, is

it reasonable to assume that a large SV - say a duplication of 50 kb - that includes one TE of 1kb, was transposon-mediated? Can we rule out the possibility that the TE played no role in driving such a duplication? The authors need to include in the manuscript some discussion of how confidently we can infer that SVs were truly mediated by TEs.

3. Expression of RPLP2 pseudogenes

Section 2.3 reports that some RPLP pseudogenes are expressed at different levels in high and low altitude populations. However, the authors need to report in the manuscript:

- How many individuals were used to represent each population in the RNA-seq data
- Whether these data were from the same tissue type and life stage for all individuals
- How consistent the expression patterns are among individuals from the same population and/or whether the differences in expression level between populations is statistically significant.

Without this information, it is impossible to evaluate the biological relevance of this finding.

4. Role of RPLP2 pseudogenes in adaptation

The authors hypothesise that expression of some RPLP2 pseudogenes might contribute to adaptation to different altitudes in *P. glacialis* (Discussion). However, as far as I can tell, they did not investigate whether these pseudogenes are associated with increased *F_{st}* between high and low altitudes, which would be expected under their hypothesis. I recommend that they test for this pattern.

5. LINEs and RPLP2 pseudogenes

The authors hypothesise that LINE retrotransposons could be responsible for the proliferation of RPLP2 pseudogenes in *P. glacialis* (Discussion). Is this a testable hypothesis? Could the authors test for an association between the pseudogenes and LINE elements in the genome? This would greatly increase the support for their hypothesis.

6. "Selective sweep genes"

The authors use the term "selective sweep genes" in the Abstract, Introduction, Results, Discussion and Methods. For example, Section 5.5 states "In order to identify the selective sweep genes between the high- and low- altitude populations".

This language is oversimplifying a complex problem. First, the authors are identifying signatures consistent with historical selective sweeps, but which could be caused by other things (e.g. background selection or stochastic noise). Secondly, signatures of selective sweeps usually span broad genomic regions, and the exact target of selection is unknown (and sometimes it is not a gene at all). I therefore advise that the authors use more nuanced language that communicates the uncertainty in these analyses. For example "In order to identify signatures of selective sweeps, and associate these with genes that might be the targets of selection ...". Finally, the authors report 473 windows with signatures consistent with sweeps, but they say that only 12 genes were associated (Section 2.5). They need to explain how this conclusion was reached. Were some candidate sweep regions not close to any gene? Was some distance threshold used to determine whether a gene was associated with a putative sweep? This needs to be explained in the manuscript.

7. Selective sweep signatures

Section 5.5: "Based on the top 1% of the *P_i* (BQTM/BQXL) and *F_{ST}* values, altitude-related selective

sweep genes of *P. glacialis* were identified.”

The authors used the π ratio to identify a subset of F_{st} outliers that are more likely to have experienced selective sweeps. In the Methods section, they should explain the logic of this approach, and also explain why the cut-offs of 0.36 and 1.38 for the π ratio were used. Furthermore, I would strongly recommend using additional approaches that are better at distinguishing selective sweeps from background selection and random noise, such as Tajima’s D or Fay and Wu’s H , or Sweepfinder and other similar tools.

8. Environmental stress and TE content

The Discussion states “During the cyclonic glacial and interglacial periods of QCA, these *Parnassius* species (like *P. glacialis* or *P. apollo*, etc) spreading out of the QTP had to face more pressure of climate change due to the loss of their nearby natural refuge of the vertical altitude gradient on the QTP.

Therefore, the *P. glacialis* and *P. apollo* species exhibit more TE insertions than the *P. orleans* species on the QTP.”

This statement is implying causality, where causality has not been proved. I recommend replacing the final sentence with a more cautious interpretation, like “The increased environmental stress experienced by *P. glacialis* and *P. apollo* might explain the increased TE content in their genomes.”

9. Genome size comparisons

Am I correct that the genome size comparisons were performed by comparing assembly length for two species (*P. orleans*, *P. apollo*) and inferred genome size from Illumina reads using the GCE software (*P. glacialis*, *P. acdestis*, *P. simo*, and *P. nomion*)? If so, it is important to ensure that estimates from these different methods are comparable in this genus.

10. Figure 3a - the authors must please explain in the figure caption what the numbers alongside the tree mean (e.g. “+703 / -2062 1 1 434”).

11. Section 2.2: “These findings suggest that the recombination rate (solo-LTRs/complete LTRs) of LTR retrotransposons has significantly influenced the genome sizes of *Parnassius* butterflies.”

This statement needs more explanation. I assume the authors are referring to the idea that non-homologous recombination (also called ectopic or unequal recombination) between LTRs can reduce genome size and leave solo-LTRs as a footprint, but this needs to be clarified in the manuscript.

12. Section 2.4: The use of the term “branch” to describe the results of the Admixture analysis (Alexander et al. 2009) is not accurate. This software models each individual as a product of admixture between k ancestral populations (also called source populations or clusters). I recommend rewording statements like “another branch included the remaining eight populations” to something like “individuals from the remaining eight populations were assigned to the second cluster”, which more accurately captures the underlying logic of the analysis.

13. Section 5.4: “The divergence time from TimeTree (<http://www.timetree.org/>) served as the reference”

There should be one or more citable papers whose data is provided by TimeTree. It is important to cite these, rather than just citing TimeTree, as the data on TimeTree will change as more studies are conducted.

14. Section 5.4: RPLP genes were identified using a process of Tblastn, with Exonerate and Genewise. Why only for this family of genes? Was this first identified as a family of interest using the automated annotation and then these were chosen for further analysis? The decision to focus specifically on this gene family for detailed analysis should be explained in the manuscript.

15. Section 5.4: "genes of was"

16. Section 5.5: "further SNP filtering using the option '--indep-pairwise 10 kb 10 0.5'"
Please explain what this does and the purpose of it.

17. Section 5.5 "gene flow was examined with Treemix v1.13"
Please state whether this used the complete SNP set or the LD-pruned one.

Signed: Simon Martin

Reviewer #2 (Remarks to the Author):

Generally the manuscript is well written. The data is impressive, though some aspects are not presented with sufficient clarity. Also, the analyses are generally well done.

My biggest concerns are on clarity of the presented data and the wording in the discussion.

In several places (e.g. Lines 307-310 on demography, 310-314 on TE activity) the wording is overreaching on what the data can conclusively show. Here the authors make statements where their interpretation is rather stated as fact. One example is also in the conclusion "Additionally, the transposon-mediated genetic differentiation facilitates the acquisition of new phenotypic characteristics (e.g., body size) and local adaptation for *P. glacialis*." The word "facilitates" suggests that the link between transposons and local adaptation is a given, though in reality this remains a purely correlative hypothesis at best. I feel the results are intriguing enough in themselves, but the language should be clear on what is speculation and what is fact. The interpretation regarding the functions of individual genes (Lines 315-328) is worded as hypothetical but these are insufficiently argued and read more like speculation. I understand this is generally a difficulty with population genomic studies, where functional interpretation is limited to correlative analyses. But the authors may revisit the wording here.

Also, the data could be presented more clearly. For example, some data are also not shown, e.g. size distribution data for the different species. Was this measured or taken from the literature? This is also true regarding the population genomic data. More information should be given, for instance the sequencing coverage, read length, if it is pair-ends as well as what quality filtering was done, and final data quality. In the text there are several spots where it is unclear why analyses are performed on which species. The analyses range from addressing one species intro) or three species (line 107, Fig. 2) to six

species (line 82). This is confusing and the rationale should be outlined early on, e.g. at end of introduction.

Additional comments

Line 82: At this point it remains unclear which 6 species you mean and how this data was generated or collated. Be more specific.

Lines 94f: The genomic data is not referenced via accession numbers or similar.

Line 121: Please cite the rationale for recombination rate.

Line 183: Considering the importance of genome size in the remaining paper it would be interesting to see genome size estimates of the *P. glacialis* specimens from the nine populations, pending that resequencing depth is sufficient

Line 201f: What are the numbers for the intermediate populations?

Lines 454: How were the altitude-related selective sweep genes identified?

Response to comments from Reviewer 1:

Comments: Reviewer report for the revised version of “Out of the Qinghai-Tibet Plateau: genome evolution and potential local adaptation of alpine butterflies *Parnassius glacialis* driven by transposable elements”

Youjie Zhao and co-authors investigated genome evolution and population history of *Parnassius glacialis*, a butterfly with an exceptionally large genome that is thought to have expanded its range outward from the Qinghai-Tibet Plateau. They compare genome size and content between *Parnassius* species, describe population structure and differentiation between various *P. glacialis* populations at different altitudes, and discuss evidence that TEs play a role in local adaptation. This is an interesting study system for investigating genome evolution and local adaptation. The authors have generated a powerful dataset, and provided a comprehensive description of the genome and population genomic patterns, which would make this work of interest to others interested in genome size evolution and local adaptation. However, I believe that some of the claims in this paper, including the claim in the title that local adaptation is “driven extensively by transposable elements”, have not been adequately proved, because the authors do not consider and test alternative explanations for the observations. Below I describe in detail what I mean by this, and suggest ways the authors could improve the support for their claims. I also highlight several areas where the manuscript provides insufficient detail to describe analyses or interpretations. I believe all of these concerns can be addressed with some additional analyses and modifications to the text.

Response

Thanks very much for taking your time to review our manuscript. We really appreciate all your valuable comments and suggestions! We have carefully considered these suggestions and tried our best to improve the manuscript. Following your constructive comments, we added some additional analyses and corresponding modifications to the text. We hope that the revised manuscript can address all of the concerns. Please find our itemized responses in the blue part.

Thanks again!

Comment 1. local adaptation

The authors find that SV regions overlapping TEs, have higher differentiation (F_{st}) between altitudes on average. They state: “These findings indicate that different types of transposons might have varying effects on accelerating genetic differentiation in *P. glacialis*.” (Section 2.4) “TE activity may drive genetic differentiation between high- and low-altitude *P. glacialis* populations.” (Section 3) This hypothesis is plausible, but it is not the only possible explanation for the observed results. It is well known that regions of lower recombination rate tend to have higher differentiation due to lower effective population size (e.g. Burri et al. Evolution Letter 2017 <https://doi.org/10.1002/evl3.14>). Therefore, an alternative interpretation of these results is that TEs might be more common in regions of low recombination rate. In other words, TEs might not be causal of the increased differentiation, but rather a consequence of lower recombination rate. The authors need to discuss this alternative interpretation of the results. In order to claim that TEs drive local adaptation, I would further recommend testing the alternative interpretation by examining estimated recombination rates. For example, see how this was tested by Montejo-Kovacevich et al. 2022 Nat. Comm. <https://doi.org/10.1038/s41467-022-32316-x> (Figure 3c).

Response to comment 1:

We are grateful to you for the suggestion about the recombination rate. Following this comment, we calculated the recombination rate (per bp) of high- and low- altitude populations (BQXL and BQTM) based on the tool LDhelmet as previous studies (Martin et al., 2019, Montejo-Kovacevich et al., 2022). We found that the average recombination rate of high-latitude population (BQXL) was higher than that of low-altitude population (BQTM). Furthermore, we analyzed the relationship between recombination rate, TE-SVs and F_{ST} in *P. glacialis*. As you said, the recombination rate showed negative correlation ($p < 0.01$) with F_{ST} value. Given the coevolution between TE and recombination in previous study (Kent et al, 2017), we speculated that they were likely to cooperatively influence genetic differentiation in *P. glacialis*. Based on the additional analysis, we revised the manuscript as follows.

Line 522-523 in method section 5.7:

“The recombination rates (per bp) of high- and low- altitude populations (BQXL and BQTM) were calculated by the tool LDhelmet¹¹³ as previous studies^{114,115}.”

Line 242-248 in result section 2.4:

“Additionally, the average recombination rate (0.128) of BQXL population was significantly higher than that (0.061) of BQTM population, which showed on every chromosome in *P. glacialis* (Additional file 1: Fig. S13a). The regions of No TE-SVs were found to have higher recombination rates (0.081 and 0.148) than TE-SVs’ (0.049 and 0.118) in both BQTM and BQXL populations (Additional file 1: Fig. S13b). Furthermore, the recombination rate showed negative correlation ($p < 0.01$) with F_{ST} value in *P. glacialis* (Additional file 1: Fig. S13c).”

Line 336-338 in discussion:

“Previous studies have shown that lower recombination could also lead to higher differentiation⁵³. Given the coevolution between TE and recombination in previous study⁵⁴, it is likely that they both contribute to influence the genetic differentiation in *P. glacialis*.”

Comment 2. Transposon-mediated SVs

Section 5.6: “Regions where SVs overlapped with TEs were annotated as transposon-mediated SVs” Please provide further detail here. Was it necessary that the SV was equivalent in size to the TE? If not, is it reasonable to assume that a large SV - say a duplication of 50 kb - that includes one TE of 1kb, was transposon-mediated? Can we rule out the possibility that the TE played no role in driving such a duplication? The authors need to include in the manuscript some discussion of how confidently we can infer that SVs were truly mediated by TEs.

Response to comment 2:

We appreciate and agree with your suggestion that stricter parameters are needed to set for the identification of TE-mediated SVs. Following this advice, we tried different parameters of SVs overlapped with TEs ranged from 50% to 100%, and found that the filtered SVs accounted for 75% to 52% of the total SVs, respectively Considering the sequence variation of TE-mediated SVs, we selected the SVs overlapped with TEs for more than 90% as TE-SVs. Based on the new parameters, we revised the Method and Result sections in the manuscript as follows.

Line 520-521 in the method section 5.7:

“Regions where SVs overlapped with TEs for more than 90% were annotated as TE-SVs, overlap of >0 to 90% as uncertain, and 0 overlap as No TE-SVs.”

Line 233-236 in the result section 2.4:

“A total of 201,680 structural variations (SVs) were identified between BQXL and BQTM populations. Of these, 140,666 (70%) SVs were overlapped with TEs for more than 90%, and this substantial number of transposon-mediated SVs was primarily distributed across 80,891 10-kb windows in the *P. glacialis* genome.”

Comment 3. Expression of RPLP2 pseudogenes

Section 2.3 reports that some RPLP pseudogenes are expressed at different levels in high and low altitude populations. However, the authors need to report in the manuscript:

- How many individuals were used to represent each population in the RNA-seq data
- Whether these data were from the same tissue type and life stage for all individuals
- How consistent the expression patterns are among individuals from the same population and/or whether the differences in expression level between populations is statistically significant.

Without this information, it is impossible to evaluate the biological relevance of this finding.

Response to comment 3:

We thank you for these suggestions. We have added the sample information of RNA data into the additional file 2: Table-S3, that is, three samples came from the same life stage (adult) and same tissue (thorax) for each population of BQXL and BQTM. Meanwhile, we have conducted the differential expression analysis of RNA-seq for 434 RPLP2 genes by the tool edgeR ($p < 0.05$), and the results showed that one RPLP2 gene (*Pglac-RPLP2-19*) was significantly up-regulated in the BQTM populations. Then we added the data of differential expression into the additional file 2: Table-S4 in the revised the manuscript as follows.

Line 427-428 in method section 5.3:

“Transcriptome sequences of 6 adult individuals (muscle isolated from thorax) from two *P. glacialis* populations (BQXL and BQTM) were downloaded from the NCBI SRA database (Additional file 2: Table S3)”

Line 478 in method section 5.4:

“Additionally, edgeR ($p < 0.05$) was used to identify the RPLP2 genes with differential expression⁹⁸.”

Line 189-195 in result section 2.3:

“ Among these processed pseudogenes with normal expression, total 4 pseudogenes (*Pglac-RPLP2-1*, *Pglac-RPLP2-2*, *Pglac-RPLP2-3* and *Pglac-RPLP2-19*) showed specific expression in the low-altitude population BQTM (Fig. 3g), among them, 3 (*Pglac-RPLP2-1*, *Pglac-RPLP2-2*, *Pglac-RPLP2-3*) were clustered in the Clade1 branch of RPLP2 phylogenic tree (Additional file1: Fig. S9a), *Pglac-RPLP2-19* was the only one of significantly up-regulated RPLP2 gene compared to BQXL population (Additional file1: Fig. S9c, additional file2: Table S4).”

Line 310-312 in discussion section:

“It is particularly interesting to note that there are several specific expressed pseudogenes (*Pglac-RPLP2-1*, *Pglac-RPLP2-2*, *Pglac-RPLP2-3* and *Pglac-RPLP2-19*) with start codons and one of them (*Pglac-RPLP2-19*) was significantly up-regulated in the low altitude population BQTM (Fig. S9c, Additional file 2: Table S4, Additional file 12)”

Comment 4. Role of RPLP2 pseudogenes in adaptation

The authors hypothesise that expression of some RPLP2 pseudogenes might contribute to adaptation to different altitudes in *P. glacialis* (Discussion). However, as far as I can tell, they did not investigate whether these pseudogenes are associated with increased F_{ST} between high and low altitudes, which would be expected under their hypothesis. I recommend that they test for this pattern.

Response to comment 4:

Thank you for the suggestion. Following this comment, we analyzed the F_{ST} of 434 RPLP2 genes between high- and low-altitude populations by vcfTools (with the operation: --weir-fst-pop [BQTM] --weir-fst-pop [BQXL] --chr [chromosome] --from-bp [start-hit] --to-bp [end-hit]). The results showed that most RPLP2 genes (378) have the invalid F_{ST} values (0) due to the lack of effective SNP sites in these regions. One possible reason for this is the short length of RPLP2 pseudogenes. According to the evaluation of SNP density (Genome size/SNP number), the average length per SNP was about 138 bp in *P. glacialis*. We found that most RPLP2 pseudogenes are less than 300 bp in length, which means that there are only 2 SNP sites per gene. Another reason is that most SNPs filtered from populations were generally distributed in the low repetitive regions. Massive duplications also result fewer effective SNP sites for these RPLP2 pseudogenes in *P. glacialis*. Although we could not obtain the valid F_{ST} values, the available evidence (including RPLP2 expansion, higher K_a/K_s , and RNA expression) for RPLP2 pseudogenes still suggests that they might have more opportunities to evolve into new functional genes as a response to some environmental stresses. Finally, we modified the description about pseudogene adaptation in the Discussion as follows.

Line 315-318 in discussion section:

We changed the sentence:

“we hypothesize that these RPLP2 processed pseudogenes have undergone rapid evolution to obtain the ability of functional genes, which help *P. glacialis* to cope with challenges in low-altitude environments outside the QTP”

To:

“we hypothesize that these RPLP2 processed pseudogenes have undergone rapid evolution to obtain the ability of functional genes, possibly as a response of *P. glacialis* to the low-altitude environments outside the QTP”

Comment 5. LINES and RPLP2 pseudogenes

The authors hypothesise that LINE retrotransposons could be responsible for the proliferation of RPLP2 pseudogenes in *P. glacialis* (Discussion). Is this a testable hypothesis? Could the authors test for an association between the pseudogenes and LINE elements in the genome? This would

greatly increase the support for their hypothesis.

Response to comment 5:

We agree with your opinion that more evidences are needed for the association between TE and RPLP2 pseudogenes. Earlier studies demonstrated that processed pseudogenes were resulted from retrotransposition mediated by LINE transposons (Zhang et al., 2002; Troskie et al., 2021) or LTR retrotransposons (Jamain et al., 2001; Tan et al., 2016). LINE-mediated pseudogenes usually harbor the hallmark sequences of poly(A) tail and target site duplications (TSDs) in their flanking regions (Terai et al., 2010; Richardson et al., 2014). Although many ribosomal pseudogenes have been identified as the LINE-mediated productions in human, we cannot find the signs of a poly(A) tail or TSDs in the flanking regions of RPLP2 genes in *P. glacialis*. Instead of it, we found new evidence about the association between LTR retrotransposon and RPLP2 pseudogene in this study. LTR-mediated gene duplications have the hallmark repeated LTRs in their flanking regions, which formed a chimera consisting of pseudogene and LTR retrotransposon (Jamain et al., 2001; Tan et al., 2016). For these RPLP2 pseudogenes, we designed a pipeline to identify the putative LTR-mediated duplications (Additional file 1: Fig. S7), and the results showed that most of the RPLP2 pseudogenes were likely located in a gypsy family of LTR retrotransposon, forming the similar chimeras (Additional file 1: Fig. S8). Based on this additional analysis, we re-wrote the sentences in the revised manuscript as the following.

Line 480-490 in the method section 5.5:

“5.5 Identified the association between RPLP2 pseudogene and LTR retrotransposon

Gene duplication mediated by LTR retrotransposon has the hallmark repeats of LTR retrotransposon in their flanking regions⁴⁹. Firstly, the 8kb flanking sequences of each RPLP2 gene were aligned to check for this repeat by BLAT⁹⁹. Secondly, the host LTR retrotransposons inserted by RPLP2 pseudogene were identified stepwise in Additional file 1: Fig. S7. In the analysis, two major gypsy families (Hic_asm_15-Gypsy-8551066-1339-5650 and Hic_asm_0-Gypsy-28602735-1211-5694) with the open reading frame (ORF) of group-specific antigen (GAG) and polymerase (POL) were found to associate with the RPLP2 pseudogenes in *P. glacialis* (Additional file 14). Considering the high similarity (97.8%) of reverse transcriptase domain (RT)¹⁰⁰ between the two families (Additional file 15), we finally used the longer one (Hic_asm_15-Gypsy-8551066-1339-5650) to identify the structure of LTR-mediated RPLP2 pseudogenes by RepeatMasker v4.0.6 (-nolow -cutoff 600)⁶⁷ (Additional file 1: Fig. S7, S8, additional file 11).”

Line 173-177 in the result section 2.3:

“It is worth noting that most of the RPLP2 pseudogenes (381) were found to be likely located in a gypsy family (Hic_asm_15-Gypsy-8551066-1339-5650), forming the similar chimeras with the structure of 5'-LTR, pseudogene, INTERNAL and 3'-LTR (Additional file 1: Fig. S7, Fig. S8, additional file 11). We also found that 5'-LTRs were generally shorter than 3'-LTRs in these chimeras (Additional file 1: Fig. S8, Additional file 11).”

Line 301-305 in the discussion section:

“Earlier studies demonstrated that processed pseudogenes typically result from retrotransposition mediated by LINE transposons⁴⁸ or LTR retrotransposons⁴⁹. Considering that most of the RPLP2

pseudogenes were likely located in the region of LTR retrotransposons (Additional file 1: Fig. S8, additional file 11), we hypothesize that the RPLP2 gene family generates numerous processed pseudogenes through these retrotransposons (Fig. 3a, 3b).”

Comment 6. “Selective sweep genes”

The authors use the term “selective sweep genes” in the Abstract, Introduction, Results, Discussion and Methods. For example, Section 5.5 states “In order to identify the selective sweep genes between the high- and low- altitude populations”. This language is oversimplifying a complex problem. First, the authors are identifying signatures consistent with historical selective sweeps, but which could be caused by other things (e.g. background selection or stochastic noise). Secondly, signatures of selective sweeps usually span broad genomic regions, and the exact target of selection is unknown (and sometimes it is not a gene at all). I therefore advise that the authors use more nuanced language that communicates the uncertainty in these analyses. For example “In order to identify signatures of selective sweeps, and associate these with genes that might be the targets of selection ...”. Finally, the authors report 473 windows with signatures consistent with sweeps, but they say that only 12 genes were associated (Section 2.5). They need to explain how this conclusion was reached. Were some candidate sweep regions not close to any gene? Was some distance threshold used to determine whether a gene was associated with a putative sweep? This needs to be explained in the manuscript.

Response to comment 6:

This is a useful suggestion for us. Following this comment, we changed the description about “Selective sweep genes” in the manuscript. Meanwhile, based on the methods of P_i and F_{ST} , we added the additional analysis of XPEHH for the selective sweeps. The candidate regions (10 kb windows) were identified by the bottom 1% of P_i ratios (BQTM/BQXL), top 1% of F_{ST} values and top 1% of XPEHH values. Subsequently, the corresponding genes covered at least 50% of these regions were selected for the subsequent annotation. Accordingly, we revised the manuscript as follows.

Line 30 in the abstract section:

We changed “selective sweep genes” to “genes with selective signatures”.

Line 271 in the Figure 5 of result section:

We changed “Selective sweep genes” to “Selective signatures”.

Line 340 in the discussion section:

We changed “selective sweep genes” to “genes with selective signatures”.

Line 505-511 in the method section: 5.6

“In order to identify the signatures of selective sweep, three methods (P_i , F_{ST} and XPEHH) were used for the high- and low- altitude populations (BQXL and BQTM). PopgenWindows (https://github.com/simonhmartin/genomics_general/) was employed to calculate the P_i and F_{ST} values for each 10-kb non-overlapping window containing at least 10 SNPs. XPEHH statistics were calculated for each 10-kb window using selscan¹⁰⁸. Based on the bottom 1% of P_i ratios (BQTM/BQXL), top 1% of F_{ST} values and top 1% of XPEHH values, the candidate regions of selective sweep were identified for the low-altitude population (BQTM). Finally, the corresponding

genes covered at least 50% in these regions were selected for the subsequent annotation with the databases of NCBI and SwissProt⁸³.”

Line 258-261 in the result section 2.5:

“The F_{ST} value for each window was calculated, and from these, 1,223 regions (approximately top 1%) with the highest F_{ST} values ($F_{ST} \geq 0.1854$) were selected (Additional file 13). 243 of these 1,223 regions with the lowest Pi ratios (BQTM/BQXL ≤ 0.3578 , bottom 1%) and highest XPEHH values ($XPEHH \geq 2.330$, top 1%) were identified as potential selective sweep regions (Fig. 5b).”

Line 262-264 in the result section 2.5:

“Genome annotation revealed that the 243 selected regions for the BQTM population were primarily overlapped with 38 genes, and 9 of these genes were covered at least 50 % by the regions of selective sweeps (Fig. 5a, Table S6)”

Comment 7. Selective sweep signatures

Section 5.5: “Based on the top 1% of the Pi (BQTM/BQXL) and F_{ST} values, altitude-related selective sweep genes of *P. glacialis* were identified.”

The authors used the pi ratio to identify a subset of F_{ST} outliers that are more likely to have experienced selective sweeps. In the Methods section, they should explain the logic of this approach, and also explain why the cut-offs of 0.36 and 1.38 for the pi ratio were used. Furthermore, I would strongly recommend using additional approaches that are better at distinguishing selective sweeps from background selection and random noise, such as Tajima’s D or Fay and Wu’s H , or Sweepfinder and other similar tools.

Response to comment 7:

Thank you for the suggestion. In previous version, we used Pi and F_{ST} method to identify the selective sweeps. The bottom 1% (<0.36) and top 1% (>1.38) of Pi ratios (BQTM/BQXL) represent the selective signatures of BQTM and BQXL, respectively. Here, we added the additional analysis of XPEHH for the selective sweeps. The candidate regions (10 kb windows) for the low altitude population (BQTM) were identified by the bottom 1% of Pi ratios (BQTM/BQXL), top 1% of F_{ST} values and top 1% of XPEHH values. Accordingly, we updated the manuscript as follows.

Line 505-511 in the method section: 5.6

“In order to identify the signatures of selective sweep, three methods (Pi , F_{ST} and XPEHH) were used for the high- and low- altitude populations (BQXL and BQTM). PopgenWindows (https://github.com/simonhmartin/genomics_general/) was employed to calculate the Pi and F_{ST} values for each 10-kb non-overlapping window containing at least 10 SNPs. XPEHH statistics were calculated for each 10-kb window using selscan¹⁰⁸. Based on the bottom 1% of Pi ratios (BQTM/BQXL), top 1% of F_{ST} values and top 1% of XPEHH values, the candidate regions of selective sweep were identified for the low-altitude population (BQTM). Finally, the corresponding genes covered at least 50% in these regions were selected for the subsequent annotation with the databases of NCBI and SwissProt⁸³.”

Comment 8. Environmental stress and TE content

The Discussion states “During the cyclonic glacial and interglacial periods of QCA, these *Parnassius* species (like *P. glacialis* or *P. apollo*, etc) spreading out of the QTP had to face more pressure of climate change due to the loss of their nearby natural refuge of the vertical altitude gradient on the QTP. Therefore, the *P. glacialis* and *P. apollo* species exhibit more TE insertions than the *P. orleans* species on the QTP.” This statement is implying causality, where causality has not been proved. I recommend replacing the final sentence with a more cautious interpretation, like “The increased environmental stress experienced by *P. glacialis* and *P. apollo* might explain the increased TE content in their genomes.”

Response to comment 8:

As suggested by your comment, we have revised the manuscript as follows.

Line 284-286 in discussion:

“Therefore, the increased environmental stress experienced by *P. glacialis* and *P. apollo* might explain the increased TE content in their genomes.”

Comment 9. Genome size comparisons

Am I correct that the genome size comparisons were performed by comparing assembly length for two species (*P. orleans*, *P. apollo*) and inferred genome size from Illumina reads using the GCE software (*P. glacialis*, *P. acdestis*, *P. simo*, and *P. nomion*)? If so, it is important to ensure that estimates from these different methods are comparable in this genus.

Response to comment 9:

You are right. In the previous version, the genome size of *P. orleans* and *P. apollo* came from the literature, and the other 4 species were newly assessed in our study. Now, to avoid assessment errors caused by different methods, we re-assessed the genome size of 6 *Parnassius* species using same methods. Accordingly, we updated the manuscript as follows.

Line 392-396 in method section 5.2:

“To assess genome size, we downloaded the genome sequencing data (Illumina sequencer) for five *Parnassius* species (*P. acdestis*, *P. simo*, *P. orleans*, *P. apollo* and *P. nomion*)⁵⁵ from NCBI SRA database (Additional file 2: Table S1). For *P. glacialis*, we used the sequencing data from Illumina and PacBio sequencers for assessment, respectively (Additional file 2: Table S1). Jellyfish 2.2.10⁵⁶ and GCE software v1.0.0 were used to estimate the genome sizes of these six *Parnassius* species based on their respective 17 k-mer frequency distributions (Additional file 1: Fig. S1)⁵⁷.”

Line 84-91 in result section 2.1:

“genome sizes of six representative *Parnassius* species (*P. acdestis*, *P. simo*, *P. orleans*, *P. nomion*, *P. apollo* and *P. glacialis*) at varying elevations from 300 to 5,000 meters a.s.l. were evaluated using genome sequencing (Additional file 2: Table S1). The assessment results indicated that the genome sizes of the six *Parnassius* species ranged from 1.0 to 1.40 Gb (Fig. 1, additional file 1: Fig. S1), and species at low/median elevations possessed relatively larger genome sizes compared to those at high elevations (Fig. 1). For *P. glacialis*, the genome size was estimated to be approximately 1.33 Gb and 1.35 Gb based on the Illumina and PacBio reads (Additional file 1: Fig. S1).”

Comment 10. Figure 3a - the authors must please explain in the figure caption what the numbers alongside the tree mean (e.g. “+703 / -2062 1 1 434”).

Response to comment 10:

Thank you for the suggestion. We have added the explanation for the Figure 3a.

Line 150-151 in result section 2.3:

“Fam (+/-) represents the number of expansion and contraction of gene families in eight butterfly species. RPLP0, RPLP1 and RPLP2 represent the number of each family in eight butterfly species.”

Comment 11. Section 2.2: “These findings suggest that the recombination rate (solo-LTRs/complete LTRs) of LTR retrotransposons has significantly influenced the genome sizes of Parnassius butterflies.”

This statement needs more explanation. I assume the authors are referring to the idea that non-homologous recombination (also called ectopic or unequal recombination) between LTRs can reduce genome size and leave solo-LTRs as a footprint, but this needs to be clarified in the manuscript.

Response to comment 11:

Following your suggestion, we made according changes in the manuscript.

Line 127, 290 and 292 in section 2.2 and section 3

We changed the words from “recombination rate” to “non-homologous recombination rate”.

Comment 12. Section 2.4: The use of the term “branch” to describe the results of the Admixture analysis (Alexander et al. 2009) is not accurate. This software models each individual as a product of admixture between k ancestral populations (also called source populations or clusters). I recommend rewording statements like “another branch included the remaining eight populations” to something like “individuals from the remaining eight populations were assigned to the second cluster”, which more accurately captures the underlying logic of the analysis.

Response to comment 12:

Following this comment, we have made the changes in the manuscript.

Line 203-204 in result section 2.4:

We changed the sentence from “another branch included the remaining eight populations” to “individuals from the remaining eight populations were assigned to the second cluster” in the Section 2.4.

Comment 13. Section 5.4: “The divergence time from TimeTree (<http://www.timetree.org/>) served as the reference”

There should be one or more citable papers whose data is provided by TimeTree. It is important to cite these, rather than just citing TimeTree, as the data on TimeTree will change as more studies are conducted.

Response to comment 13:

Thank you for the suggestion. We have added the citable papers about the divergence time.

Line 448-449 in Section 5.4:

“Considering that there were fewer fossils of butterflies, we selected two calibration points based on previous studies ^{5,90}: (1) the crown of *Parnassius* (10.5–16.6 Ma) ⁵; (2) the crown of Papilionoidea (110.3–86.9 Ma) ⁹⁰.”

Comment 14. Section 5.4: RPLP genes were identified using a process of Tblastn, with Exonerate and Genewise. Why only for this family of genes? Was this first identified as a family of interest using the automated annotation and then these were chosen for further analysis? The decision to focus specifically on this gene family for detailed analysis should be explained in the manuscript.
RPLP

Response to comment 14:

Thank you for the suggestion. We have added the explanation for focusing on this gene family in the manuscript.

Line 454-456 in method section 5.4:

“The initial results of Café analysis showed that *P. glacialis* harbored a significant expanded group containing 17 RPLP genes, while the other species had only one or two. At the same time, we found some RPLP genes without introns (like pseudogene) in *P. glacialis*.”

Comment 15. Section 5.4: “genes of was”

Response to comment 15:

Thank you for the suggestion. We changed the words from “the divergence time of RPLP2 genes of was” to “the divergence time of RPLP2 genes was” in line 471 of section 5.4.

Comment 16. Section 5.5: “further SNP filtering using the option ‘--indep-pairwise 10 kb 10 0.5’”
Please explain what this does and the purpose of it.

Response to comment 16:

Thank you for pointing this out. We added the explanation about the options “the options (window size 10 kb, step size 10kb, and threshold 0.5) of linkage disequilibrium (LD)” in line 501 of section 5.6.

Comment 17. Section 5.5 “gene flow was examined with Treemix v1.13”

Please state whether this used the complete SNP set or the LD-pruned one.

Response to comment 17:

Thank you for pointing this out. We used the LD-pruned SNP set which described in line 501 of section 5.6.

“the options (window size 10 kb, step size 10kb, and threshold 0.5) of linkage disequilibrium (LD)”

Response to comments from Reviewer 2:

Comments: Reviewer report for the revised version of “Out of the Qinghai-Tibet Plateau: genome evolution and potential local adaptation of alpine butterflies *Parnassius glacialis* driven by transposable elements”

Generally the manuscript is well written. The data is impressive, though some aspects are not presented with sufficient clarity. Also, the analyses are generally well done.

Response:

Thanks very much for your careful review and valuable comments on my manuscript. We have carefully considered these helpful comments and tried our best to improve the manuscript. We hope that the revised manuscript can address all of the concerns. Please find our itemized responses in the blue part. Thanks again!

Comment 1

My biggest concerns are on clarity of the presented data and the wording in the discussion.

In several places (e.g. Lines 307-310 on demography, 310-314 on TE activity) the wording is overreaching on what the data can conclusively show. Here the authors make statements where their interpretation is rather stated as fact.

Response to comment 1

We are grateful for the suggestion. Following this comment, we have added a more detailed interpretation regarding the discussion in the manuscript as follows.

Line 330-332 on demography in discussion:

“This finding suggests that *P. glacialis* dispersed into low-altitude areas during the cold period of late QIA based on ancestral geographic reconstruction¹⁸, and subsequently underwent a genetic bottleneck after the LGM, with only a small number of adapted individuals surviving in these regions (Fig. 4c, 4d).”

Line 333-338 on TE activity in discussion:

“Concurrently, genome regions with transposon-mediated SVs were found to harbor lower recombination rate and higher F_{ST} values than those without (Fig. 4e, 4f, Fig. S13b), indicating that TE activity probably tended to accelerate the genetic differentiation between high- and low-altitude *P. glacialis* populations. Previous studies have shown that lower recombination could also lead to higher differentiation⁵³. Given the coevolution between TE and recombination in previous study⁵⁴, it is likely that they both contribute to influence the genetic differentiation in *P. glacialis*.”

Comment 2

One example is also in the conclusion “Additionally, the transposon-mediated genetic differentiation facilitates the acquisition of new phenotypic characteristics (e.g., body size) and local adaptation for *P. glacialis*.” The word “facilitates” suggests that the link between transposons and local adaptation is a given, though in reality this remains a purely correlative hypothesis at best. I feel the results are intriguing enough in themselves, but the language should be clear on what is speculation and what is fact.

Response to comment 2:

Thanks for your suggestion. We updated the sentence in the revised manuscript as the following.

Line 362-363 in conclusion:

“Additionally, the transposon-mediated genetic differentiation probably provided an opportunity for selective sweeps and potential local adaptation of *P. glacialis*.”

Comment 3

The interpretation regarding the functions of individual genes (Lines 315-328) is worded as hypothetical but these are insufficiently argued and read more like speculation. I understand this is generally a difficulty with population genomic studies, where functional interpretation is limited to correlative analyses. But the authors may revisit the wording here.

Response to comment 3

Thanks for your suggestion. We updated the discussion about these genes with selective signatures in the revised manuscript as follows.

Line 340-355 in discussion:

“Furthermore, a series of genes with selective signatures were identified for the low-altitude BQTM population (Fig. 5, Table S6), involving functions in development and immunity. For example, as a member of larval cuticle protein (LCP) gene family, LCP17 (Fig. 5a) was shown to affect body development in *Bombyx mori*³². Considering that *B. mori* is also a member of Lepidoptera, the selective signature of LCP17 in the oxygen-rich environment at low altitudes might be related to the body size enlargement for *P. glacialis* in these regions. Another example is *rc3h1* as a multifunctional regulator of immune homeostasis, found to mediate the STING-dependent innate immune response in *Drosophila melanogaster*³⁰; its selective signature in *P. glacialis* of low altitudes (Fig. 5a) is also interpreted as likely related to enhancing the immune defense against increased pathogenic agents in humid and warm climates in these areas. Other selective signatures, including *Anpep* (also known as APN, a receptor in resistance to insecticidal crystal (Cry) toxins in Lepidoptera)³¹, *Sad* (a member of Cytochrome P450 enzymes)³⁵, *Bcr* (a Rho GTPase regulator)³⁸ and *twk-18* (a member of TWiK family)³⁶ have been found to be associated with detoxification metabolism, ecdysteroid biosynthesis and nervous system in previous studies (Table S6). Together, we suggest that these genes probably reflected the potential adaptive evolution of *P. glacialis* associated with low-altitude environments, such as enriched oxygen content, warmer climate and more active pathogenic microorganisms.”

Comment 4

Also, the data could be presented more clearly. For example, some data are also not shown, e.g. size distribution data for the different species. Was this measured or taken from the literature?

Response to comment 4

Many thanks for your suggestion. We mainly used the body size data of *P. glacialis* at different altitudes in our study. The body sizes of 10 *P. glacialis* populations have been measured in previous study of our lab (the 18th cited reference in our manuscript).

Hao, X., Mao, Z., Ren, H. & Rao, R. Analysis of geometric morphological of vein of *Parnassius glacialis* in different geographic populations. *J. Anhui Agric. Sci.* 34, 84–88 (2017).

Comment 5

This is also true regarding the population genomic data. More information should be given, for instance the sequencing coverage, read length, if it is pair-ends as well as what quality filtering was done, and final data quality.

Response to comment 5

Thank you for the suggestion. We added the information of genomic data in the revised manuscript as follows.

Line 493-494 in method section 5.6:

“We obtained a total of ~ 489 Gb clean reads from 25 newly sequencing individuals at ~19.5 Gb (14X) per individual (Additional file 2: Table S5).”

Line 380-382 in method section 5.1:

“Long-read libraries with a fragment size of 20 kb were constructed using the SMRTbell Template Prep Kit (Pacific Biosciences), while 150 bp paired-end libraries with an insert size of 350 bp were constructed using the TruSeq Nano DNA Library Prep Kit (Illumina). The 20 kb and 150 bp paired-end libraries were sequenced using the PacBio HiFi and Illumina HiSeq X Ten instruments, respectively (Additional file 2: Table S1, S5).”

Line 385-388 in method section 5.1:

“The raw reads of Illumina sequencer were filtered to remove reads with adaptor, low-quality reads and duplicated reads using FastQC (<https://github.com/s-andrews/FastQC/>). The QC procedures were as follows: (a) removal of reads with $\geq 10\%$ unidentified nucleotides (N), (b) removal of reads with $> 20\%$ of bases with a Phred quality < 5 , (c) removal of reads with > 10 nucleotides aligned to the adaptor, allowing $\leq 10\%$ mismatches, and (d) removal of putative PCR duplicates generated by PCR amplification during the library construction process.”

Comment 6

In the text there are several spots where it is unclear why analyses are performed on which species. The analyses range from addressing one species (intro) or three species (line 107, Fig. 2) to six species (line 82). This is confusing and the rationale should be outlined early on, e.g. at end of introduction.

Response to comment 6

Thank you for the suggestion. Following this comment, we revised the introduction of the manuscript as follows.

Line 73-80 at end of introduction:

“In this study, we assessed the genome size variation among six *Parnassius* species at different altitudes. Subsequently, we newly assembled the chromosome-level genome of *P. glacialis* and explored the role of transposable elements (TEs) in driving the evolution of its large genome through

comparative analysis with the reported genomes of *P. orleans*¹⁵ and *P. apollo*¹⁶. Based on the genome sequencing of 41 individuals from 9 *P. glacialis* populations at different altitudes ranging from 300 to 1,800 m a.s.l., we analyzed the genetic structure and explored the impact of TEs on the genetic differentiation for these populations. This study will help us to understand the mechanisms of genome evolution and potential local adaptation for *P. glacialis* butterflies.”

Comment 7

Additional comments

Line 82: At this point it remains unclear which 6 species you mean and how this data was generated or collated. Be more specific.

Response to comment 7:

Thank you for the suggestion. We added the sample information of 6 species in additional file 2: Table S1. And we updated the revised manuscript as follows.

Line 369-371 in method section 5.1:

“Two 5th instar larvae of *P. glacialis* were collected from an altitude of 300 m a.s.l. in Laoshan, Nanjing, China. One larva was starved for 48 h and then rapidly frozen in liquid nitrogen until it was used for genome survey and the *de novo* genome sequencing. The other larva was used for Hi-C sequencing (Additional file 2: Table S1).

Line 392-394 in method section 5.2:

“To assess genome size, we downloaded the genome sequencing data (Illumina sequencer) for five *Parnassius* species (*P. acestis*, *P. simo*, *P. orleans*, *P. apollo* and *P. nomion*)⁵⁵ from NCBI SRA database (Additional file 2: Table S1). For *P. glacialis*, we used the sequencing data from Illumina and PacBio sequencers for assessment, respectively (Additional file 2: Table S1).”

Comment 8

Lines 94f: The genomic data is not referenced via accession numbers or similar.

Response to comment 8:

Thanks for your suggestion. We've submitted the genomic data to the NCBI database and got the accession number (SRR24653388).

Comment 9

Line 121: Please cite the rationale for recombination rate.

Response to comment 9:

Thanks for your suggestion. We added the citation (Devos et al., 2002) about the recombination of LTR retrotransposons.

Line 127 in Section 2.2:

“These findings suggest that the non-homologous recombination rate²⁶ (solo-LTRs/complete LTRs) of LTR retrotransposons has significantly influenced the genome sizes of *Parnassius* butterflies.”

Comment 10

Line 183: Considering the importance of genome size in the remaining paper it would be interesting

to see genome size estimates of the *P. glacialis* specimens from the nine populations, pending that resequencing depth is sufficient

Response to comment 10

Thanks for your suggestion. Following this comment, we tried to assess the genome size of nine populations. We merged the sequences of three individuals for each population, and assessed the genome sizes based on 17 Kmer frequency (Kmer_number/Kmer_depth) obtained from Jellyfish (like Additional file 1: Fig. S1). The results showed that the genome sizes of nine populations for adult individuals were ranged from 1.55 to 1.70 Gb, a little higher than the 1.35 Gb of chromosome-level assembly for the larval. One possible explanation is that the genome of larvae treated with hunger has less pollution than that of the adult. Another explanation is that the real heterozygous genome may be larger than 1.35 Gb, with only one haploid genotype anchoring the chromosome during genome assembly. However, it does not affect our main results: *Parnassius* species at low elevations possessed relatively larger genome sizes compared to those at high elevations. Furthermore, we have added another analysis of genome collinearity with *Papilio bianor* to check the *P. glacialis* genome (we did not select the genomes of *P. orleans* and *P. apollo* because they could not reach the chromosome level). Accordingly, we revised the manuscript as follows.

Line 405-406 in method section 5.2:

“According to the gene annotations of *P. glacialis* and *Pa. bianor*²⁴, the chromosome collinearity was constructed using the tools JCVI v1.3.4 (Additional file 1: Fig. S3)⁶²”

Line 98-99 in result section 2.1:

“Additionally, the genome collinearity revealed that 29 chromosomes of *P. glacialis* were completely mapped to 30 chromosomes of the *Papilio bianor*²⁴ butterfly (Additional file 1: Fig. S3).”

Comment 11

Line 201f: What are the numbers for the intermediate populations?

Response to comment 11:

Thanks for your suggestion. The P_i values of intermediate populations (BQLJ, BQTT, and BQTA) were approximately 0.00180, 0.00170, and 0.00166, respectively. We added these values in the Fig. 4c.

Comment 12

Lines 454: How were the altitude-related selective sweep genes identified?

Response to comment 12:

Thanks for your suggestion. We used three methods (P_i , F_{ST} and XPEHH) to identify the selective sweep signatures. And we updated the description in the method and result section.

Line 505-511 in the method section: 5.6

“In order to identify the signatures of selective sweep, three methods (P_i , F_{ST} and XPEHH) were used for the high- and low- altitude populations (BQXL and BQTM). PopgenWindows

(https://github.com/simonhmartin/genomics_general/) was employed to calculate the P_i and F_{ST} values for each 10-kb non-overlapping window containing at least 10 SNPs. XPEHH statistics were calculated for each 10-kb window using selscan¹⁰⁸. Based on the bottom 1% of P_i ratios (BQTM/BQXL), top 1% of F_{ST} values and top 1% of XPEHH values, the candidate regions of selective sweep were identified for the low-altitude population (BQTM). Finally, the corresponding genes covered at least 50% in these regions were selected for the subsequent annotation with the databases of NCBI and SwissProt⁸³.”

Line 258-261 in the result section 2.5:

“The F_{ST} value for each window was calculated, and from these, 1,223 regions (approximately top 1%) with the highest F_{ST} values ($F_{ST} \geq 0.1854$) were selected (Additional file 13). 243 of these 1,223 regions with the lowest P_i ratios (BQTM/BQXL ≤ 0.3578 , bottom 1%) and highest XPEHH values (XPEHH ≥ 2.330 , top 1%) were identified as potential selective sweep regions (Fig. 5b).”

Line 262-264 in the result section 2.5:

“Genome annotation revealed that the 243 selected regions for the BQTM population were primarily overlapped with 38 genes, and 9 of these genes were covered at least 50 % by the regions of selective sweeps (Fig. 5a, Table S6)”

REVIEWERS' COMMENTS

Reviewer #1 (Remarks to the Author):

The authors have extensively revised their manuscript and performed a number of new analyses. They have done a thorough job addressing my concerns, and I now feel that the claims made are in line with the evidence provided.

I have a few remaining minor comments that should be easily addressed.

1. Figure 3e. Check spelling in x-axis

2. Section 2.4. Use of the word “branch” to describe the Admixture results is incorrect. One instance was corrected in response to my first comments, but there are several other uses of “branch”. Terms like “cluster” or “source population” should be used instead.

3. New recombination rate results

I'm pleased to see that the authors have analysed recombination rate, which sheds light on why TE-mediated SVs may have higher F_{st} values.

The authors need to be specific in stating that LDHelmet estimates the population recombination rate (also called ρ) which is equivalent to $4N_e r$, where r is the per-base crossover recombination rate. A higher value of ρ in BQXL than BQTM could either be caused by higher crossover recombination rate (r), or larger effective population size (N_e). These cannot be distinguished without a direct measure of recombination (i.e. linkage mapping). Given that r usually evolves quite slowly, it seems more likely that the difference in ρ is caused by a difference in N_e . This caveat should be acknowledged in the paper.

4. Discussion: “lower unequal recombination rate of LTR retrotransposons in *P. glacialis* and *P. apollo* at relatively low altitudes (Fig. 2c) could lead to increased TE content, potentially explaining the formation of larger genome sizes in these *Parnassius* species spreading out of the QTP”

I understand that unequal recombination can reduce LTR content by creating solo-LTRs, but does it also reduce the content of LINE elements (which are also larger in the low-altitude species according to Figure 2b)? If not, I suggest stating that unequal recombination could be “contributing to” the difference in genome size, rather than “explaining” it.

Reviewer #2 (Remarks to the Author):

In the previous version of the manuscript my main concern was that the analyses and results did not support the strong statements made in the discussion. The authors have done a very careful job of addressing these comments as well as those of reviewer 1. Overall the manuscript is much improved and the interpretations are now better supported by the data and the updated analyses.

Response to comments from Reviewer 1:

Comments:

The authors have extensively revised their manuscript and performed a number of new analyses. They have done a thorough job addressing my concerns, and I now feel that the claims made are in line with the evidence provided. I have a few remaining minor comments that should be easily addressed.

Response:

Thanks very much for your careful review on my manuscript. Following your useful suggestions, we have carefully revised the manuscript. We hope that the revised manuscript can address all of the concerns. Please find our itemized responses in the blue and red part. Thanks again!

Comment 1. Figure 3e. Check spelling in x-axis

Response to comment 1:

We revised the spelling error “sepcies” to “species” in x-axis of Figure 3e.

Comment 2. Section 2.4. Use of the word “branch” to describe the Admixture results is incorrect. One instance was corrected in response to my first comments, but there are several other uses of “branch”. Terms like “cluster” or “source population” should be used instead.

Response to comment 2:

Following your comment, we revised the Section 2.4 of manuscript as follows.

In Section 2.4 of Results:

“Concurrently, genetic structure analysis using Admixture showed (Fig. 4c) that at $k = 2$, the southeast population BQTM formed one cluster while individuals from the remaining eight populations were assigned to the second cluster; at $k = 3$, three clusters were supported, with one cluster from $k = 2$ containing the southeast population BQTM, a second cluster comprising BQLS and BQKY, and a third cluster, including the other six populations mostly at higher altitudes; at $k = 4$, the southeast populations BQTM and BQLS each formed separate clusters, a third cluster included the northeast populations BQKY and BQTA, and the fourth cluster consisted of the remaining five populations at altitudes between 600 and 1800 m. The Admixture line graph indicated that $k = 2$ was the best pattern (Fig. S11), where BQTM and the other eight populations each formed a cluster, consistent with principal component analysis results (Fig. 4b).”

Comment 3. New recombination rate results

I’m pleased to see that the authors have analysed recombination rate, which sheds light on why TE-mediated SVs may have higher F_{st} values.

The authors need to be specific in stating that LDHelmet estimates the population recombination rate (also called ρ) which is equivalent to $4N_e \cdot r$, where r is the per-base crossover recombination rate. A higher value of ρ in BQXL than BQTM could either be caused by higher crossover recombination rate (r), or larger effective population size (N_e). These cannot be distinguished without a direct measure of recombination (i.e. linkage mapping). Given that r usually evolves quite slowly, it seems more likely that the difference in ρ is caused by a difference in N_e . This caveat should be acknowledged in the paper.

Response to comment 3:

Thanks for your useful suggestion. We revised the manuscript as follows.

In Methods section:

“To analyze the relationship between TE-SV and recombination rate (ρ), we calculated the recombination rate ($\rho=4N_e*r$) of high- and low- altitude populations (BQXL and BQTM) by the tool LDhelmet¹¹³ as previous studies^{114,115}. Among them, N_e is the effective population size and r is the crossover recombination rate per generation per bp¹¹⁵.”

In Discussion section:

“Additionally, a higher value of recombination rate ρ ($=4N_e*r$) in BQXL than BQTM (Fig. S13a) could either be caused by higher crossover recombination rate (r), or larger effective population size (N_e). Given that r usually evolves quite slowly, it seems more likely that the difference in ρ is caused by a difference in N_e , which is also supported by the PSMC result (Fig. 4d).”

Comment 4. Discussion: “lower unequal recombination rate of LTR retrotransposons in *P. glacialis* and *P. apollo* at relatively low altitudes (Fig. 2c) could lead to increased TE content, potentially explaining the formation of larger genome sizes in these *Parnassius* species spreading out of the QTP”. I understand that unequal recombination can reduce LTR content by creating solo-LTRs, but does it also reduce the content of LINE elements (which are also larger in the low-altitude species according to Figure 2b)? If not, I suggest stating that unequal recombination could be “contributing to” the difference in genome size, rather than “explaining” it.

Response to comment 4:

Yes. As you mentioned, unequal recombination showed to reduce LTR content by creating solo-LTRs, but it could not reduce the content of LINE elements. Following your comment, we revised the discussion of manuscript as follows.

In Discussion section:

“In comparison with high-altitude *P. orleans* species, the lower unequal recombination rate of LTR retrotransposons in *P. glacialis* and *P. apollo* at relatively low altitudes (Fig. 2c) could lead to increased TE content, potentially **contributing to** the formation of larger genome sizes in these *Parnassius* species spreading out of the QTP (Fig. 1).”

Response to comments from Reviewer 2:

Reviewer #2 (Remarks to the Author):

In the previous version of the manuscript my main concern was that the analyses and results did not support the strong statements made in the discussion. The authors have done a very careful job of addressing these comments as well as those of reviewer 1. Overall the manuscript is much improved and the interpretations are now better supported by the data and the updated analyses.

Response

Thanks very much for taking your time to review our manuscript.